# Descending GABAergic pathway links brain sugar-sensing to peripheral nociceptive gating in *Drosophila*

Mami Nakamizo-Dojo ®[1,3], Kenichi Ishii ®[1,3], Jiro Yoshino[1,3], Masato Tsuji[1] & Kazuo Emoto ®[1,2] ✉

Although painful stimuli elicit defensive responses including escape behavior for survival, starved animals often prioritize feeding over escape even in a noxious environment. This behavioral priority is typically mediated by suppression of noxious inputs through descending control in the brain, yet underlying molecular and cellular mechanisms are incompletely understood. Here we identify a cluster of GABAergic neurons in *Drosophila* larval brain, designated as SEZ-localized Descending GABAergic neurons (SDGs), that project descending axons onto the axon terminals of the peripheral nociceptive neurons and prevent presynaptic activity through GABA_B receptors. Remarkably, glucose feeding to starved larvae causes sustained activation of SDGs through glucose-sensing neurons and subsequent insulin signaling in SDGs, which attenuates nociception and thereby suppresses escape behavior in response to multiple noxious stimuli. These findings illustrate a neural mechanism by which sugar sensing neurons in the brain engages descending GABAergic neurons in nociceptive gating to achieve hierarchical interaction between feeding and escape behavior.

Animals typically display one behavior at a time, and thus need to assign behavioral priorities[1,2]. This behavioral priority is influenced by an animal's behavioral states as well as its internal states. One remarkable example is the behavioral conflict between nociceptive responses and feeding. Once an animal receives potentially harmful stimuli such as harsh mechanical inputs, pungent chemicals, or noxious heat, it immediately escapes from noxious environments[3]. In food-deprived animals, however, feeding often takes precedence over escape responses. For instance, food-deprived cats are less likely to withdraw from acute noxious cutaneous heat during eating[4], and food-deprived chickens show fewer pain-motivated behavior in response to tonic pain produced in the legs while eating takes place[5]. Likewise, medicinal leeches show reduced nociceptive response to harsh chemicals and electric shock while feeding[6,7]. Prioritization of feeding over nociceptive responses is likewise observed in crayfish and mollusks[8–10].

In addition, sugar feeding produces analgesic effects in rat and human infants[11,12].

Previous studies suggest that the hierarchical interaction between feeding and nociceptive responses is in part mediated by nociceptive suppression through descending control in the brain[13–15]. Noxious stimuli are typically received by nociceptive neurons, the sensory neurons specialized for their detection[16–18]. In mammals, these noxious inputs are relayed directly to spinal motor neurons to provoke escape behavior and to the second-order neurons in the spinal cord to higher-order processing centers in the brain[15,19,20]. This nociceptive sensory processing is often modulated by both the local interneurons in the spinal cord and the descending neurons in the brain that directly innervate the nociceptive circuit in the spinal cord[15,21]. In rodents, electrophysiological studies showed that sugar feeding to starved animals upregulates neural activity of a subpopulation of descending

[1]Department of Biological Sciences, Graduate School of Science, The University of Tokyo, 7-3-1 Hongo, Bunkyo-ku, Tokyo 113-0033, Japan. [2]International Research Center for Neurointelligence (WPI-IRCN), 7-3-1 Hongo, Bunkyo-ku, Tokyo 113-0033, Japan. [3]These authors contributed equally: Mami Nakamizo-Dojo, Kenichi Ishii, Jiro Yoshino. ✉e-mail: emoto@bs.s.u-tokyo.ac.jp

neurons in the brainstem and that pharmacological inactivation of the brainstem neurons attenuates the pain suppression by sugar feeding[22,23]. These data suggest that certain types of descending neurons in the brain are involved in pain suppression during feeding, yet detailed regulatory mechanisms including how feeding engages descending control in pain suppression remain elusive.

Recent studies reveal that *Drosophila* larvae have a nociceptive sensory system with a circuit arrangement partially analogous to that in mammals[24–26]. Noxious stimuli elicit stereotyped escape behaviors in *Drosophila* larvae including rolling followed by fast crawling[27–29]. This noxious stimulus-evoked escape behavior is triggered by peripheral nociceptive neurons, the class IV da (C4da) neurons, which densely innervate the body wall with their dendritic branches and project their axons to the ventral nerve cord (VNC) (Fig. 1a)[30–33]. In the VNC, C4da axon terminals form synaptic contacts with multiple identified second-order neurons including some with ascending projections to the brain (for example, A08n) and others, such as mCSIs, that output to

motoneurons for reflex responses[27,28,34,35]. Like nociceptive circuits in mammals, the nociceptive circuit in *Drosophila* larvae receives modulatory inputs from local neurons in the VNC[34,35]. However, to date, descending inputs that modulate nociceptive processing and behavior have not been identified in *Drosophila* larvae.

In this study, we have identified a small cluster of GABAergic neurons in the *Drosophila* larval brain, designated as SEZ-localized Descending GABAergic neurons (SDGs), that project descending axons onto axonal terminals of C4da neurons in the VNC. Calcium imaging and optogenetic manipulation revealed that SDGs negatively regulate C4da synaptic activity through GABA_B receptors, which terminates escape rolling behavior. Furthermore, we found that feeding D-glucose, but not L-glucose and arabinose, significantly attenuated noxious stimuli-evoked escape behavior in starved larvae. This glucose-induced nociceptive suppression is mediated by the glucose-sensing CN neurons[36] that directly evoked insulin release from the insulin-producing cells (IPCs) in the brain upon sugar refeeding, which

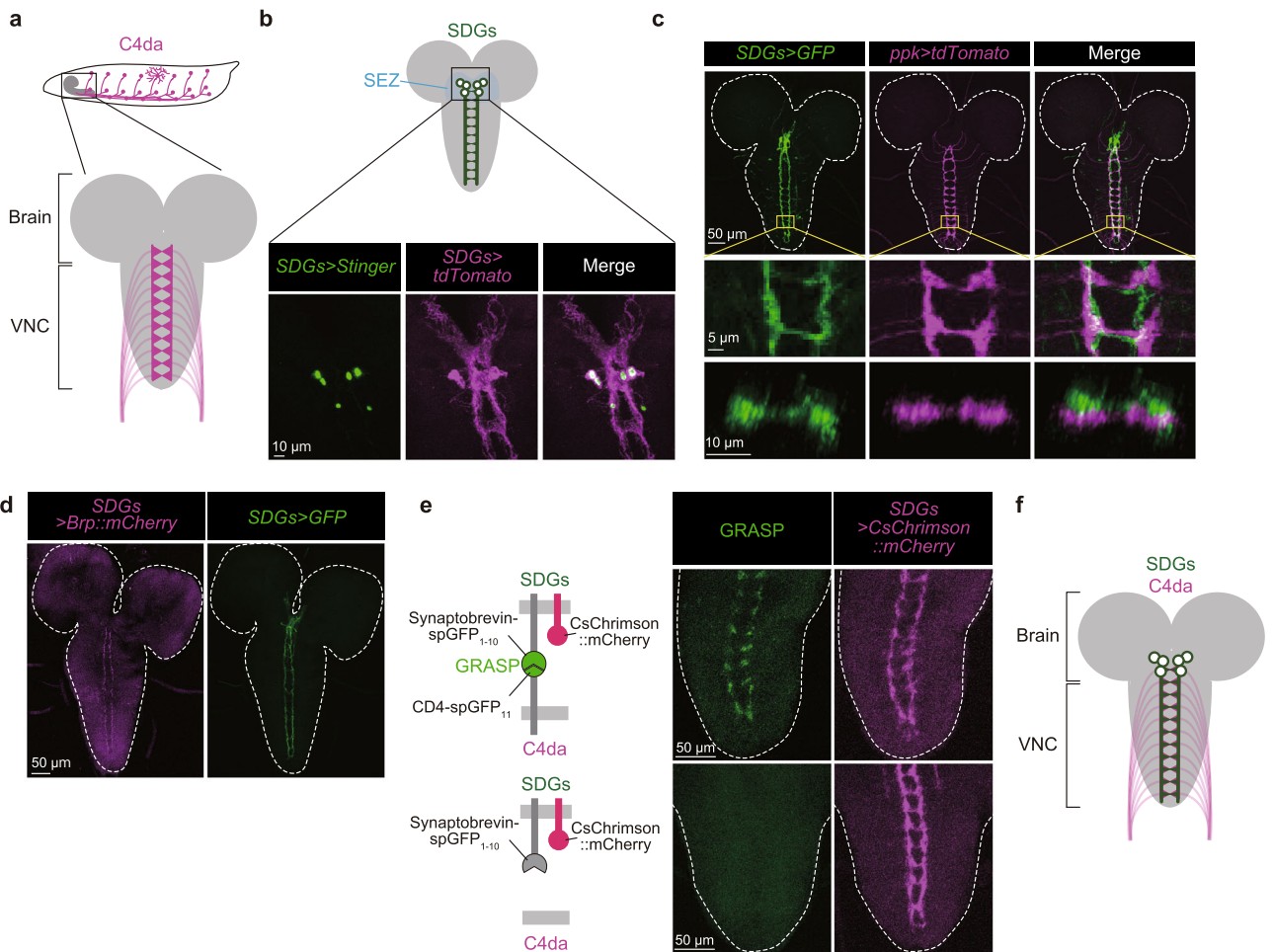

**Fig. 1 | Identification of the descending SDGs that innervate C4da axon terminals. a** Top, a schematic lateral view of C4da neurons in third instar larvae. Bottom, a schematic ventral view of axon terminal projections of C4da neurons in the ventral nerve cord (VNC). **b** Cell bodies of SDGs located in the larval brain. Nuclear-targeted GFP (green) and membrane-bound tdTomato (magenta) were driven by *SDGs-spGAL4*. Similar results were obtained across 3 independent samples per genotype. **c** Dual labeling of SDGs expressing mCD8::GFP (green) and C4da neurons visualized with *ppk-CD4::tdTomato* (magenta). Top panels show the signals in the larval brain and VNC. Middle panels show magnified images of the abdominal A6–7 segments in the VNC (the yellow squared area in the top panels). Bottom panels are transverse

sections of the A7 segment. Similar results were obtained across 5 independent samples per genotype. **d** Expression of the presynaptic marker Brp::mCherry (magenta), and mCD8::GFP (green) under the control of *SDGs-spGAL4*. Similar results were obtained across 5 independent samples per genotype. **e** Left, a schematic view of Syb-GRASP signals between SDGs and C4da neurons. Middle, confocal images of Syb-GRASP signals in the larval VNC. Right, the axon terminals of SDGs are labeled by CsChrimson::mCherry. Note that each pair of images placed side-by-side shows an identical region. Similar results were obtained across 4 independent samples per genotype. See Supplementary Table 1 for full genotypes. **f** A schematic image of synaptic contacts from SDGs to C4da axon terminals.

promoted sustained SDGs activation via insulin receptors and thereby attenuates nociception. These findings support a model in which sugar feeding suppresses nociception through glucose sensing neurons and subsequent insulin signaling in GABAergic descending neurons, ensuring hierarchical interactions between feeding and escape behavior.

## Results

### Identification of SDGs, a previously uncharacterized sub-population of descending neurons innervating axon terminals of C4da nociceptive neurons

To isolate descending neurons that modulate larval escape behavior, we searched for neurons that project to the regions of the neuropil occupied by C4da terminals (Fig. 1a). A visual screen of 6849 Janelia GAL4 collections (https://flweb.janelia.org/cgi-bin/flew.cgi) revealed 9 lines that appeared to label neurites in the VNC with close proximity to the distinctive ladder-like axon terminals of C4da neurons. Among them, R16C06-GAL4 labeled 30–40 cells that were clustered around the SEZ of the larval central brain and had descending projections partially overlapping with C4da axon terminals (Supplementary Fig. 1a). To identify the C4da-innervating descending neurons with higher cellular resolution, we searched for split-GAL4 combinations that cover the R16C06-GAL4/LexA-labeled subpopulations and found that the intersection between R21F01-p65.AD and R93B07-GAL4.DBD specifically captured 6 of the 30–40 neurons labeled by R16C06-LexA (Supplementary Fig. 1b). All 6 neurons had their cell bodies in the SEZ (Fig. 1b, c), and their descending projections exhibited a substantial overlap with the ladder-like axonal terminals of C4da neurons (Fig. 1c). Based on these characteristics, we designated this distinct sub-population as "SEZ-localized Descending GABAergic neurons (SDGs)", and henceforth refer to this combination of split-GAL4 as SDGs-spGAL4. Since the presynaptic marker Brp::mCherry expressed in SDGs was largely confined in their descending neurites in the VNC (Fig. 1d), SDGs likely project axons to the VNC. To examine whether SDGs might form synaptic contacts with C4da axon terminals, we performed activity-dependent GFP Reconstitution Across Synaptic Partners (Syb-GRASP) analysis[37] and found that presynaptic expression of one half of the split-GFP protein (spGFP$_{1-10}$) in SDGs generated a ladder-like GRASP signal along their entire axons when the other half (spGFP$_{11}$) was expressed in C4da neurons (Fig. 1e). It is thus likely that SDGs have direct synaptic contacts with C4da axon terminals (Fig. 1f).

### SDGs are necessary and sufficient to suppress escape behavior evoked by multimodal noxious stimuli

Next, to examine whether SDGs modulate escape behavior, we hyperpolarized SDGs via expression of the inwardly rectifying potassium channel Kir2.1 while optogenetically stimulating C4da neurons using the red-shifted channelrhodopsin ReaChR. We found that Kir2.1expression by either R16C06-GAL4 (Supplementary Fig. 2a) or SDGs-spGAL4 (Fig. 2a; No-GAL4 control, 1.5 (1.0–2.1) s, n = 14; No-UAS control, 1.8 (1.5–3.0) s, n = 26; SDGs silencing, 12± (7.8–15) s, n = 25; each value indicates median (interquartile range: Q1–Q3) s; Supplementary Fig. 2b and Supplementary Movie 1) significantly prolonged rolling duration compared to control. We further performed optogenetic stimulation of C4da neurons with varying light intensities and found that SDGs-silenced larvae showed significantly higher rolling probabilities compared to genetic controls at all intensities tested, even under dim light conditions where the control group failed to show any response (Fig. 2b). These data suggest that SDGs negatively regulate behavioral sensitivity to noxious stimuli as well as duration of rolling escape behavior.

We further examined whether SDGs are required for control of escape responses evoked by harsh mechanical stimuli and noxious heat, two types of noxious stimuli that activate C4da neurons to elicit nociceptive escape behaviors[28,34,35]. Consistent with the optogenetic experiments, expression of Kir2.1 in SDGs significantly shortened the rolling latency following thermo-nociceptive stimulus application through a 46 °C heat probe[38] (Fig. 2c; No-GAL4 control, 3.7 (2.5–5.1) s, n = 70; No-UAS control, 4.2 (3.0–6.0) s, n = 70; SDGs silencing, 2.1 (1.4–2.9) s, n = 70). Similarly, the Kir2.1-mediated chronic silencing of SDGs significantly increased the probability of mechanically evoked larval rolling (Fig. 2d; No-GAL4 control, 56%, n = 98; No-UAS control, 52%, n = 100; SDGs silencing, 81%, n = 100; Supplementary Movie 2). Furthermore, we used the green-light gated anion channel GtACR1[39] for acute silencing of SDGs to minimize developmental perturbations. As a result, larvae expressing GtACR1 in SDGs showed a significantly higher level of rolling probability when mechanical stimuli were applied during 10 s of green-light illumination, compared to the genetic controls and the no-light group (Fig. 2e; No-GAL4 control, 47%, n = 72; No-UAS control, 45%, n = 67; No-light control, 44%, n = 66; SDGs silencing, 86%, n = 72). These results support the notion that SDGs are required for a negative control of sensitivity and duration of escape behavior evoked by multimodal noxious stimuli.

Conversely, sustained excitation of SDGs by constitutive expression of NaChBac, a bacterial depolarization-activated sodium channel[39], significantly decreased the rolling probability after mechanical stimulation (Fig. 2f; No-GAL4 control, 70%, n = 50; No-UAS control, 60%, n = 50; SDGs activation, 16%, n = 50; Supplementary Movie 3). We also examined acute actions of SDGs on nociceptive modulation by conducting optogenetic experiments with larvae expressing the red-light-sensitive opsin CsChrimson[40,41] in SDGs. When larvae were mechanically stimulated with the von Frey filaments concomitant with optogenetic SDGs stimulation, rolling probability was significantly attenuated compared to controls (Fig. 2g; No-GAL4 control, 58%, n = 50; No-UAS control, 48%, n = 50; No-light control, 50%, n = 50; SDGs activation, 24%, n = 50). By contrast, manipulation of SDGs activity had no detectable effect on baseline behavior such as locomotion (Supplementary Fig. 2c, d). These results indicate that both chronic and acute excitation of SDGs specifically suppress rolling behavior upon noxious stimuli.

We further examined whether acute SDGs activation could terminate sustained rolling behavior, which is a key regulatory step for escape behavioral transition from rolling to fast crawling[26–28]. Allyl isothiocyanate (AITC), a pungent compound found in cruciferous plants such as wasabi, elicits larval rolling by stimulating TRPA1 channels expressed at the outer body surface including C4da nociceptors[34,42,43]. Once soaked in a high concentration of AITC solution, the control larvae continuously rolled and rarely stopped (Fig. 2h and Supplementary Fig. 2e). Strikingly, larvae expressing CsChrimson in SDGs halted their rolling behaviors in response to the red-light illumination, even after the onset of intense rolling in the AITC solution (Fig. 2h; No-GAL4 control, 15%, n = 20; No-UAS control, 20%, n = 20; SDGs activation, 70%, n = 20). In addition, in a reversed procedure where larvae were first optogenetically stimulated and then treated with AITC, the prior activation of SDGs significantly delayed the onset of nociceptive rolling (Supplementary Fig. 2f), further suggesting that excitation of SDGs leads to rolling termination upon noxious stimuli. These data, together with the silencing/activation results, indicate that SDGs negatively regulate nociceptive rolling to sculp larval escape behavior.

### SDGs suppress escape behavior via GABAergic signaling

To gain insights into how SDGs suppress escape responses, we next searched for neurotransmitters expressed in SDGs. Immunohistochemistry of neurotransmitters and related markers revealed that SDGs are all GABA-positive (Fig. 3a and Supplementary Fig. 3a, b). Moreover, the fluorescence marker driven by SDGs-spGAL4 overlapped with another marker simultaneously expressed by glutamate decarboxylase 1 (Gad1; encoding the GABA synthesis enzyme)-LexA (Fig. 3a and Supplementary Fig. 3a, b). Since SDGs were immunoreactive to

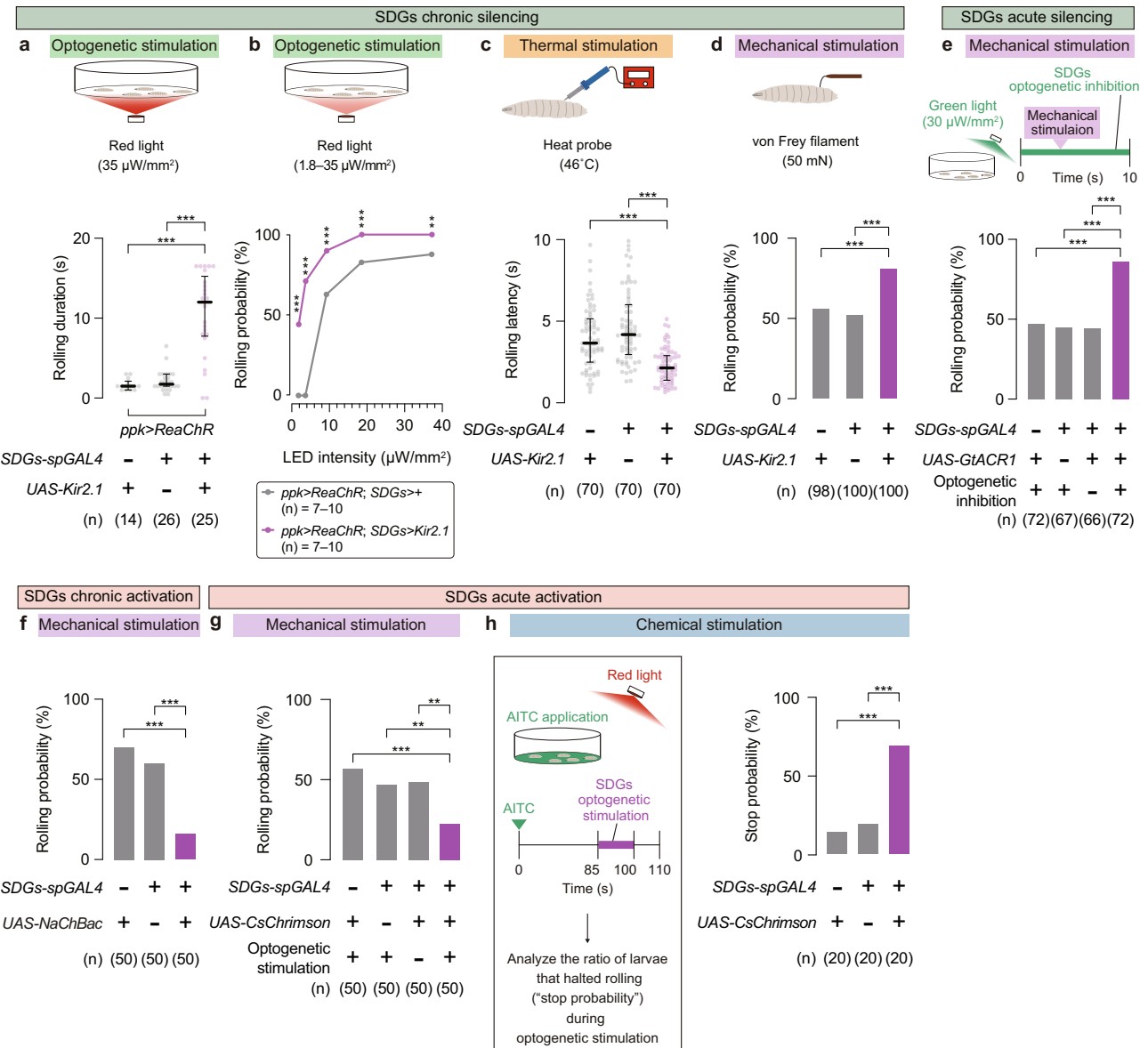

**Fig. 2 | SDGs are necessary and sufficient for suppression of C4da-mediated escape behavior. a** Rolling duration following optogenetic C4da activation. In this and following panels, 'n' indicates the number of biologically independent animals used for each group, and the thick line and the thin error bar represent the median and interquartile range, respectively. \*\*\**p* < 0.0005 (Mann–Whitney U-test with Bonferroni correction). **b** Rolling probability of SDGs-silenced larvae in which C4da neurons were optogenetically stimulated under various light intensities. Asterisks indicate statistical differences between the two genotypes at each light intensity. \*\*\**p* < 0.0005, \*\**p* < 0.005 (Fisher's exact test). **c** Effect of SDGs chronic silencing on thermo-nociceptive rolling. \*\*\**p* < 0.0005 (Mann–Whitney U-test with Bonferroni correction). **d** Effect of SDGs chronic silencing on mechano-nociceptive rolling. A local force of 50 mN was applied twice in rapid succession using the von Frey filament. Rolling probabilities after the second force application are shown. \*\*\**p* < 0.0005 (Fisher's exact test with Bonferroni correction). **e** Mechano-nociceptive rolling of larvae in which SDGs were optogenetically silenced. Top,

larvae expressing the light-gated anion channel GtACR1 in SDGs were mechanically stimulated during the 10-s window of green light illumination. Bottom, mechano-nociceptive rolling probability in larvae with or without *UAS-GtACR1* expression. \*\*\**p* < 0.0005 (Fisher's exact test with Bonferroni correction). **f** Mechano-nociceptive rolling in larvae with SDGs chronically activated via NaChBac expression. \*\*\**p* < 0.0005 (Fisher's exact test with Bonferroni correction). **g** Mechano-nociceptive rolling probability during optogenetic activation of SDGs. \*\*\**p* < 0.0005, \*\**p* < 0.005 (Fisher's exact test with Bonferroni correction). **h** Suppressive effect of SDGs optogenetic activation on AITC-elicited rolling. Left, larvae expressing CsChrimson in SDGs were treated with AITC (at 0 s) and subsequently incubated for 85 s, followed by 15 s of light stimulation. Right, probabilities for larvae already rolling in AITC to stop (distinct from "rolling probability" measured in other panels) during the 15-s optogenetic stimulation. \*\*\**p* < 0.0005 (Fisher's exact test with Bonferroni correction). See Supplementary Table 1 for full genotypes. Source data are provided as a Source Data file.

neither choline acetyltransferase (ChAT) nor vesicular glutamate transporter (VGluT) (Supplementary Fig. 3c), GABA likely serves as the major neurotransmitter in SDGs.

We next performed RNA interference (RNAi) to examine whether GABA is necessary for SDGs-dependent suppression of rolling behavior. Expression of an RNAi construct targeting *Gad1* under UAS control (*UAS-IR-Gad1*) in SDGs increased rolling probability following

mechano-nociceptive stimulation (Fig. 3b; No-*UAS-IR* control, 40%, *n* = 70; *Gad1* RNAi in SDGs, 64%, *n* = 58) while decreased rolling latency after thermo-nociceptive stimulation (Fig. 3c; No-*UAS-IR* control, 4.5 (3.0–6.7) s, *n* = 53; *Gad1* RNAi in SDGs, 1.8 (1.0–2.5) s, *n* = 48), indicating a heightened sensitivity to noxious stimuli. In addition, *Gad1* knock-down in SDGs prolonged the duration of larval rolling elicited by direct stimulation of C4da neurons with optogenetics (Fig. 3d; No-*UAS-IR*

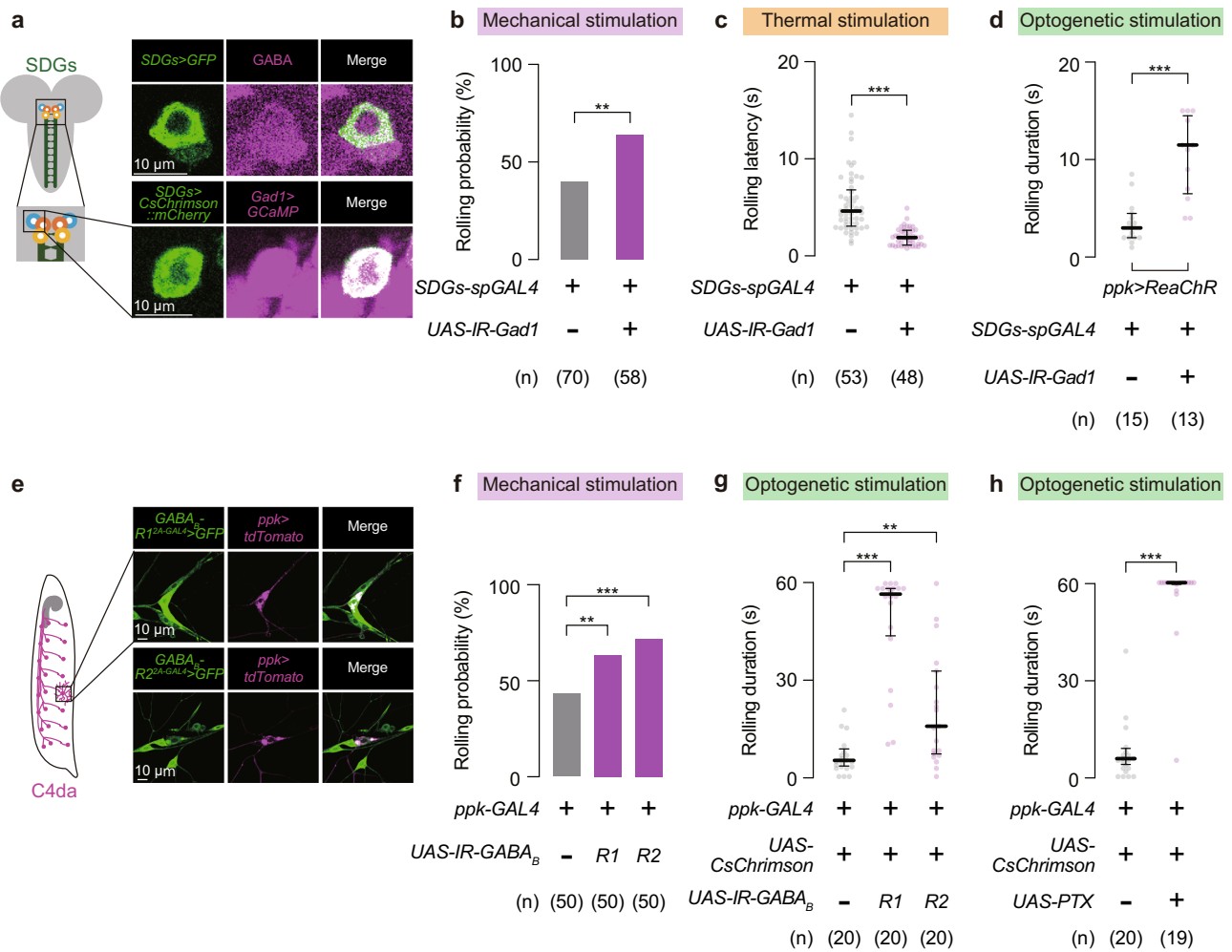

**Fig. 3 | SDGs negatively control rolling behavior through GABAergic inputs to C4da neurons. a** Top, immunohistochemistry of GABA (magenta) at the cell body of SDGs labeled with GFP (green). Bottom, dual detection of GCaMP and CsChrimson::mCherry signals driven by *Gad1-LexA* (magenta) and *SDGs-spGAL4* (green), respectively. Similar results were obtained across 5 independent samples per genotype. **b**–**d** Behavioral phenotypes caused by *Gad1* knockdown in SDGs. C4da neurons were stimulated by three distinct approaches. For mechanical stimulation (**b**), rolling probability is shown in the bar graph. Note that the background genotypes of parental flies used in **b** were different from other experiments; see Supplementary Table 1 for full genotypes. In this and following panels, 'n' indicates the number of biologically independent animals used for each group, and the thick line and the thin error bar represent the median and interquartile range, respectively. **p < 0.005 (Fisher's exact test). For thermal (**c**) and optogenetic (**d**) stimulation, latency and duration are each indicated. ***p < 0.0005 (Mann–Whitney U-test). **e** Colocalization of GFP driven by either *GABA_B-R1^{2A-GAL4}* or *GABA_B-R2^{2A-GAL4}* (green) with *ppk-CD4-tdTomato* (magenta) in C4da cell bodies. Similar results were obtained across 4 independent samples per genotype. **f, g** Nociceptive responses in larvae expressing RNAi constructs targeting each *GABA_B-R* gene in C4da neurons. Rolling probabilities after the mechanical stimulation are shown in the bar graph (**f**). ***p < 0.0005, **p < 0.005 (Fisher's exact test with Bonferroni correction). For optogenetic experiments (**g**), rolling durations within the 60-s stimulation period are indicated. ***p < 0.0005, **p < 0.005 (Mann–Whitney U-test with Bonferroni correction). **h** Effect of PTX expression on larval rolling induced by optogenetic C4da activation. ***p < 0.0005 (Mann–Whitney U-test). See Supplementary Table 1 for full genotypes. Source data are provided as a Source Data file.

control, 3.0 (2.0–4.5) s, $n = 15$; *Gad1* RNAi in SDGs, 12 (6.5–15) s, $n = 13$; and Supplementary Fig. 3d), phenocopying the effect of Kir2.1-mediated SDGs silencing on escape behavior (Fig. 2a). Similar to the *Gad1* RNAi line used above (*TRiP.HMC03350* in attP40 (BL#51794)), another two lines (*TRiP.JF02916* in attP2 (BL#28079), and *GD8508* (VDRC#32344)) induced a battery of phenotypes indicative of SDGs malfunctioning (Supplementary Fig. 3e–g).

We next shifted our focus to GABA receptors that function at the postsynaptic sites in C4da neurons. Among two classes of *Drosophila* GABA receptors, Resistant-to-dieldrin (Rdl; also known as GABA_A-R) forms a ligand-gated ion channel[44], whereas metabotropic GABA_B-Rs are G-protein-coupled receptors with three subunits (GABA_B-R1, R2, and R3) known to date[45]. Based on our GRASP data (Fig. 1e), we hypothesized that GABA receptors in C4da neurons serve as the principal postsynaptic targets of GABAergic SDGs. Labeling experiments using CRISPR/Cas9-mediated GAL4-knockins indicated that

C4da neurons express both metabotropic and ligand-gated GABA receptor genes (Fig. 3e and Supplementary Fig. 3h). To identify the GABA receptor(s) functionally relevant for larval escape behavior, we knocked down each receptor gene in C4da neurons and assayed effects on mechanically evoked nociceptive rolling. Interestingly, the rolling probability after mechanical stimulation was increased by RNAi against *GABA_B-R1* and *-R2* (Fig. 3f; No-*UAS-IR* control, 44%, $n = 50$; *GABA_B-R1* RNAi in C4da, 64%, $n = 50$; *GABA_B-R2* RNAi in C4da, 72%, $n = 50$), but not by RNAi against *GABA_B-R3* and *Rdl* (Supplementary Fig. 3i, j). Moreover, knocking down *GABA_B-R1* or *-R2* in C4da neurons significantly extended the duration of larval rolling upon C4da optogenetic activation (Fig. 3g; No-*UAS-IR* control, 5.0 (3.1–7.8) s, $n = 20$; *GABA_B-R1* RNAi in C4da, 57 (44–59) s, $n = 20$; *GABA_B-R2* RNAi in C4da, 16 (7.1–32) s, $n = 20$). These larvae also showed shortened latency to rolling induced by optogenetic C4da activation (Supplementary Fig. 3k). The above knockdown experiments targeting *GABA_B-R1*

(*TRiP.HMC03388* in attP2 (BL#51817)) and *R2* (*TRiP.HMC02975* in attP2 (BL#50608)) were phenocopied by independent RNAi elements (*KK109166* in VIE260b (VDRC#101440) for R1; *KK100020* in VIE260b (VDRC#110268) for R2) expressed in C4da neurons (Supplementary Fig. 3l, m). Furthermore, consistent with the evidence that GABA$_B$-R1 and -R2 form functional heterodimers coupled with Gα$_i$/Gα$_o$[45,46], C4da neuron-targeted expression of pertussis toxin (PTX), which inhibits Gα$_i$/Gα$_o$ proteins through selective ADP-ribosylation[47,48], effectively extended the duration of rolling induced by optogenetic C4da neuron stimulation (Fig. 3h; No-*UAS* control, 5.5 (2.3–7.8) s, $n = 20$; *PTX* in C4da, 60 (60–60) s, $n = 19$) while shortening onset latency (Supplementary Fig. 3n). These results demonstrate the requirement of heterodimeric GABA$_B$-Rs, consisting of subunits R1 and R2, to the GABAergic regulation of larval escape behavior in response to noxious stimuli.

### SDGs mediate presynaptic inhibition in C4da neurons through GABA$_B$ receptors

Given that SDGs are GABAergic neurons that likely form synaptic contacts with C4da axon terminals, we reasoned that SDGs might inhibit synaptic activity in C4da axon terminals. To test this model, we optogenetically activated C4da neurons using the channelrhodopsin-2 (ChR2[49,50]) expressed in C4da neurons and measured intracellular calcium (Ca$^{2+}$) levels at C4da axon terminals simultaneously using the red-shifted Ca$^{2+}$ indicator RGECO[33,51]. We found that the peak amplitude ($\Delta F/F_0$) during optogenetic C4da activation was significantly higher in SDGs-silenced larvae (Fig. 4a; No-*UAS* control, 76.8 (65.4–103)%, $n = 10$; SDGs silencing, 111 (92.9–115)%, $n = 10$), while significantly reduced in SDGs-activated larvae (Fig. 4b; No-*UAS* control, 295 (230–358)%, $n = 10$; SDGs activation, 153 (89.0–248)%, $n = 9$) compared to controls, indicating that SDGs negatively regulate Ca$^{2+}$ elevation in C4da axon terminals.

We further aimed to establish the role of GABA signaling in the SDGs-mediated Ca$^{2+}$ inhibition in C4da axon terminals. First as expected, bath application of GABA in an ex vivo brain preparation reduced the C4da Ca$^{2+}$ response upon optogenetic stimulation (Fig. 4c, d; without GABA, 42.7 (27.5–49.3)%, $n = 16$; GABA, 22.7 (14.6–35.4)%, $n = 19$), suggesting that GABA indeed suppresses C4da activity at axon terminals upon stimulation. Next, we applied the same genetic tools used in the behavioral assays, RNAi and PTX, to examine effects of GABA$_B$ receptors on Ca$^{2+}$ levels in C4da axon terminals and found that C4da-specific expression of either *GABA$_B$-R1* or *-R2* RNAi (Fig. 4e; No ATR, 35.5 (15.0–44.7)%, $n = 5$; No-*UAS-IR* control, 170 (153–235)%, $n = 10$; *GABA$_B$-R1* RNAi in C4da, 284 (254–313)%, $n = 9$; *GABA$_B$-R2* RNAi in C4da, 304 (224–388)%, $n = 8$) or PTX significantly increased the amplitude of Ca$^{2+}$ responses during optogenetic C4da activation (Fig. 4f; No ATR, 32.9 (23.1–49.4)%, $n = 5$; No-*UAS* control, 220 (165–245)%, $n = 10$; *PTX* in C4da, 371 (238–397)%, $n = 7$). These physiological data further support the notion that, similar to GABAergic descending neurons in the mammalian brain[15], SDGs presynaptically gate nociceptive responses of C4da neurons through a GABA signaling pathway, specifically involving the metabotropic GABA$_B$-R1 and -R2 subunits.

Since previous studies showed that GABAergic neurons are activated in response to sensory inputs and presynaptically inhibit sensory neurons though GABA$_B$ receptors in *Drosophila* olfactory and gustatory circuits[48,52–54], we hypothesized that SDGs might become physiologically activated in response to noxious stimuli, prior to exerting their inhibitory effects on peripheral nociceptors. We investigated this possibility by monitoring Ca$^{2+}$ levels of SDGs expressing the Ca$^{2+}$ indicator RGECO following optogenetic stimulation of ChR2-expressing C4da neurons. We observed a rapid surge in SDGs' Ca$^{2+}$ levels upon optogenetic C4da activation, which was absent when larvae were reared without all-*trans*-retinal (ATR), an obligate ChR2 cofactor (Fig. 4g; No ATR, 11.5 (−25.5–26.1)%, $n = 7$; ATR, 62.2 (44.0–70.8)%, $n = 7$), suggesting that C4da activation transiently

enhances SDGs activity. These findings, together with the SDGs-silencing and -activation experiments, suggest a negative feedback circuit for larval nociceptive outputs: nociceptive stimuli on peripheral C4da neurons lead to SDGs activation, which induces SDGs to send inhibitory inputs back to C4da neurons.

### Refeeding in starved larvae suppresses nociceptive responses through SDGs-mediated presynaptic inhibition in C4da neurons

SDGs are located in the SEZ, the brain region that receives multiple sensory inputs and sends out commands to the VNC and motor neurons[55,56]. We thus hypothesized that SDGs might play a role in integrating internal as well as external information into the nociceptive circuits. Inspired by the hierarchical relationship between feeding and nociception in a wide variety of organisms[6–10], we first asked whether feeding might take precedence over escape behavior in starved larvae. We found that although starvation alone had no detectable effect on nociceptive responses to noxious heat, refeeding of fasted larvae with sucrose significantly increased the latency of thermo-nociceptive responses (Fig. 5a, b; Fed, 2.2 (1.6–2.8) s, $n = 44$; Starved, 2.1 (1.2–3.4) s, $n = 55$; Refed, 8.1 (4.5–14) s, $n = 44$). Similarly, refeeding following starvation significantly reduced rolling probability upon mechano-nociceptive stimulation and optogenetic C4da activation (Fig. 5c; Fed, 65%, $n = 115$; Starved, 57%, $n = 116$; Refed, 34%, $n = 117$, and Fig. 5d; Fed, 57%, $n = 74$; Starved, 54%, $n = 80$; Refed, 28%, $n = 79$). These behavioral phenotypes were less likely attributed to general locomotion defects as sugar-refeeding did not affect larval crawling speed (Supplementary Fig. 4a). We thus examined whether SDGs are involved in the sugar refeeding-induced nociceptive suppression in starved larvae. We found that the effects of refeeding on nociceptive responses was largely canceled by silencing SDGs with Kir2.1 expression (Fig. 5e), suggesting that sugar refeeding-induced nociceptive suppression is mainly mediated by SDGs.

Since SDGs presynaptically suppressed C4da activity (Fig. 4), we next examined whether sugar refeeding to starved larvae might affect Ca$^{2+}$ elevation in C4da presynapses upon optogenetic stimulation (Fig. 5f). When ChR2-expressing C4da neurons were optogenetically activated in starved larvae refed with sucrose, we observed a significant reduction in the RGECO signal detected at C4da axon terminals, compared with those measured under fed or starved conditions (Fig. 5g; Fed, 286 (240–369)%, $n = 7$; Starved, 339 (266–497)%, $n = 6$; Refed, 150 (131–242)%, $n = 6$). Notably, these Ca$^{2+}$ responses of C4da neurons and nociceptive behavioral responses in each nutritional state mirror one another (Fig. 5b–d, h). Importantly, the suppressive effect of sugar refeeding on C4da activity was abrogated by SDGs-silencing (Fig. 5h; Fed, 234 (163–300)%, $n = 7$; Starved, 261 (154–331)%, $n = 7$; Refed, 221 (208–325)%, $n = 7$). These data together indicate that refeeding suppresses nociceptive responses in starved larvae through SDGs-mediated presynaptic inhibition in C4da neurons.

### Glucose-sensing CN neurons mediate refeeding-induced nociceptive suppression

The refeeding-induced nociceptive suppression required 3–6 h starvation (Supplementary Fig. 4b) and became effective 30–60 min after refeeding (Supplementary Fig. 4c), implying an involvement of nutritional/metabolic processes in refed larvae. Indeed, D-glucose and D-fructose, the monosaccharides produced from sucrose inside the insect gut[57], showed nociceptive suppression at the same concentration as sucrose (Fig. 6a, Supplementary Fig. 4c, d), whereas L-glucose and arabinose, monosaccharides that are sweet but nonnutritive to flies, showed no significant effect on thermo-nociceptive responses (Fig. 6a; Fed, 2.1 (1.5–3.2) s, $n = 62$; Starved, 2.0 (1.1–3.3) s, $n = 67$; Sucrose, 5.2 (2.9–8.6) s, $n = 67$; D-fructose, 3.9 (2.6–7.2) s, $n = 65$; D-glucose, 4.8 (3.2–7.2) s, $n = 59$; L-glucose, 1.8 (1.0–3.0) s, $n = 51$; Arabinose, 2.0 (1.0–2.7) s, $n = 62$). It is thus likely that digested sugars, rather than the sweet taste per se, trigger nociceptive suppression.

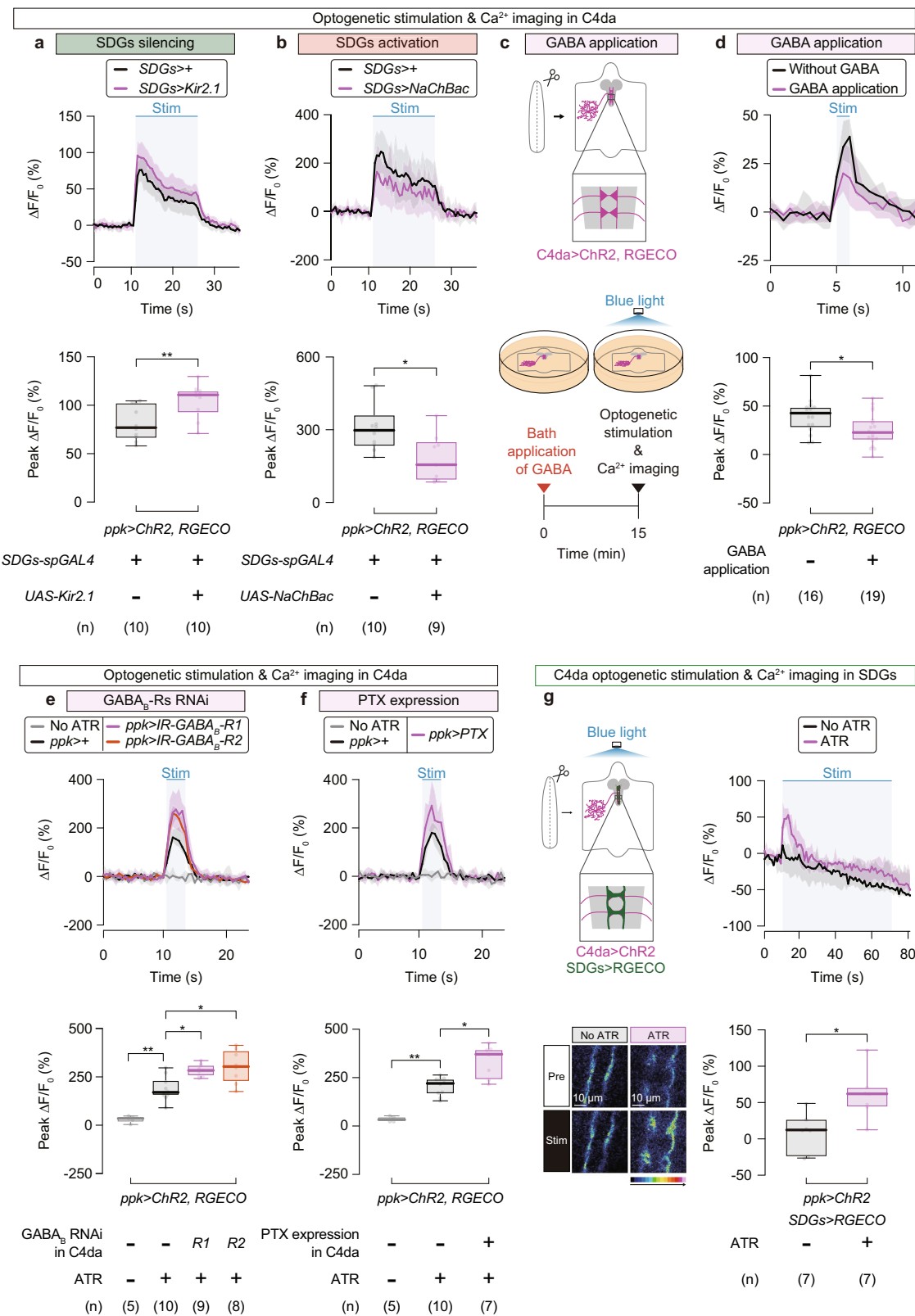

A recent study has shown that D-glucose enters a pair of corazonin (Crz)/short neuropeptide F (sNPF) double-positive neurons (named CN neurons) in the fly brain through glucose transporters and that the activated CN neurons trigger glucose-homeostasis responses[36]. Intriguingly, the monosaccharides capable of activating CN neurons, namely D-glucose and D-fructose but not L-glucose, are consistent with the types of monosaccharides capable of extending the rolling latency in refed larvae (Fig. 6a). We thus reasoned that glucose uptake and subsequent activation of CN neurons might be involved in refeeding-induced nociceptive suppression. Administration of phlorizin, a competitive inhibitor of glucose transporters that was shown to inhibit D-glucose-induced CN activation[36], significantly shortened rolling latencies in the D-glucose-refed groups to the basal level (Fig. 6b; Starved without phlorizin, 0.98 (0.64–1.6) s, $n = 41$; Starved

**Fig. 4 | SDGs mediate presynaptic inhibition of C4da neurons through GABA_B receptors. a** $Ca^{2+}$ imaging during C4da optogenetic stimulation while SDGs were silenced by Kir2.1 expression. ChR2 was activated by blue-light illumination while intracellular $Ca^{2+}$ levels were measured by the red-shifted $Ca^{2+}$ indicator RGECO. Top, traces of $\Delta F/F_0$. The thick lines (black, SDGs-spGAL4>+; magenta, SDGs-spGAL4>UAS-Kir2.1) and accompanying shades represent the median and inter-quartile range, respectively. The blue-shaded box indicates the time window of optogenetic stimulation. Bottom, peak $\Delta F/F_0$ changes. In this and following panels, 'n' indicates the number of biologically independent animals used for each group. Box plots show the median with an interquartile range, and whiskers represent the minimum-to-maximum range. **$p < 0.005$ (Mann–Whitney U-test). **b** $Ca^{2+}$ imaging of C4da while SDGs were hyperexcited via NaChBac expression. Black, SDGs-spGAL4>+; magenta, SDGs-spGAL4>UAS-NaChBac. *$p < 0.05$ (Mann–Whitney U-test). **c** Schematic design of $Ca^{2+}$ imaging in C4da. GABA was applied to the buffer

15 min prior to the photo-stimulation and imaging. **d–f** Traces of $\Delta F/F_0$ (top) and quantified changes in the peak $\Delta F/F_0$ (bottom). The effect of GABA application is shown in **d**. Black, without GABA application; magenta, with GABA application. For **e** and **f**, samples were prepared from larvae expressing either RNAi constructs for GABA_B-Rs (**e**) or PTX (**f**) in C4da. **$p < 0.005$, *$p < 0.05$ (Mann–Whitney U-test with Bonferroni correction). **g** Left, a schematic of $Ca^{2+}$ imaging in SDGs while opto-genetically stimulating C4da (top), and representative images of C4da axon terminals from larvae reared with or without ATR, prior to ("Pre") or during ("Stim") the blue-light illumination (bottom). Note that RGECO was expressed in SDGs but not in C4da, which is different from **a–f**. Right, traces of $\Delta F/F_0$ were measured in larvae reared without ATR (black) or with ATR (magenta). *$p < 0.05$ (Mann–Whitney U-test). Note that each experiment was carried out under distinct imaging condition optimized for each purpose, as detailed in Supplementary Table 2. See Supplementary Table 1 for full genotypes. Source data are provided as a Source Data file.

with phlorizin, 1.3 (0.8–1.8) s, $n = 42$; 60 mM D-glucose refed without phlorizin, 3.2 (1.8–4.5) s, $n = 45$; 60 mM D-glucose refed with phlorizin, 1.6 (0.85–2.4) s, $n = 46$; 600 mM D-glucose refed without phlorizin, 4.4 (2.9–6.5) s, $n = 47$; 600 mM D-glucose refed with phlorizin, 1.3 (0.68–1.8) s, $n = 43$; 60 mM D-fructose refed without phlorizin, 3.3 (1.7–4.3) s, $n = 43$; 60 mM D-fructose refed with phlorizin, 2.9 (1.4–5.1) s, $n = 42$; 600 mM D-fructose refed without phlorizin, 3.5 (2.6–4.8) s, $n = 40$; 600 mM D-fructose refed with phlorizin, 4.1 (2.2–6.2) s, $n = 43$. Sugar-primed activation of CN neurons in refed larvae was confirmed by increased signal intensity in calcium-dependent nuclear import of LexA (CaLexA[58]) (Supplementary Fig. 4e). To genetically test the involvement of CN neurons in the sugar-dependent nociceptive control, we performed both silencing and activation experiments. Chronic silencing of CN neurons by Kir2.1 expression overrode the delay in the rolling onset of D-glucose-refed larvae (Fig. 6c; No-GAL4 control fed, 2.6 (1.7–3.3) s, $n = 57$; No-GAL4 control starved, 2.2 (1.2–2.9) s, $n = 58$; No-GAL4 control refed, 6.7 (4.4–10) s, $n = 57$; CN silencing fed, 2.3 (1.8–3.3) s, $n = 51$; SDGs silencing starved, 2.0 (1.4–3.4) s, $n = 57$; SDGs silencing refed, 2.2 (1.6–2.8) s, $n = 51$). In addition, optogenetic inhibition of GtACR1-expressing CN neurons during the 1-h refeeding period canceled the sugar-induced rolling delay (Fig. 6d; Starved without light, 2.0 (1.5–2.8) s, $n = 23$; Refed without light, 4.0 (2.4–5.2) s, $n = 24$; Starved with light, 2.3 (1.5–3.3) s, $n = 26$; Refed with light, 1.7 (1.0–2.5) s, $n = 41$). Conversely, sustained activation of CN neurons via NaChBac expression extended the rolling latency upon thermo-nociception in normally fed larvae (Fig. 6e; No-GAL4 control, 3.0 (2.0–4.1) s, $n = 70$; No-UAS control, 2.7 (2.1–3.4) s, $n = 70$; CN activation, 4.7 (3.2–6.4) s, $n = 70$). Moreover, optogenetic stimulation of CN neurons for 1 h following the 12-h starvation (Fig. 6f) effectively suppressed the thermo-nociceptive behavior (Fig. 6g; No-UAS control, 1.8 (1.2–2.8) s, $n = 50$; CN activation, 3.2 (1.3–5.6) s, $n = 50$). This temporal CN activation without refeeding was effective against C4da responses as well; the 1-h optogenetic stimulation of CN neurons prior to imaging partly suppressed the optogenetically induced calcium influx in C4da neurons (Fig. 6h; No-UAS control, 226 (204–350)%, $n = 11$; CN activation, 124 (51.1–242)%, $n = 15$; note that CN and C4da neurons expressed different channelrhodopsins, red-light responsive CsChrimson and blue-light responsive ChR2, enabling independent stimulation). These results suggest that the glucose-sensing CN neurons mediate the sugar refeeding-induced nociceptive suppression (Fig. 6i).

### Refeeding-induced insulin signaling enhances SDGs activity and suppresses nociceptive behavior

How do CN neurons trigger SDGs-mediated nociceptive suppression? Sugar-primed CN neurons promotes secretion of insulin-like peptide 2 (Ilp2) from insulin-producing cells (IPCs) in the brain of adult flies[36]. Likewise, immunostaining with an anti-Ilp2 antibody revealed that Ilp2 signals in IPCs were significantly reduced after D-glucose refeeding (Fig. 7a), and this reduction of Ilp2 signals in refed larvae was

abrogated by silencing CN neurons (Fig. 7a; No-GAL4 control starved, 0.92 (0.74–1.2), $n = 25$; No-GAL4 control refed, 0.49 (0.18–0.79), $n = 30$; CN silencing starved, 0.76 (0.62–1.1), $n = 30$; CN silencing refed, 0.65 (0.33–0.97), $n = 22$), suggesting that refeeding in starved larvae triggers Ilp2 release from IPCs in a CN neurons-dependent manner.

We further examined whether the Ilps release from IPCs is required for refeeding-induced nociceptive suppression. To this end, we transiently blocked secretion from IPCs by expressing a temperature-sensitive form of dynamin protein (shibire^ts, abbreviated as shi^ts)[59] using insulin-like peptide 2 (ilp2)-GAL4 for temporal silencing of IPCs[36]. Larvae refed under permissive temperature showed prolonged latencies to nociceptive stimuli with no apparent differences between genotypes (Fig. 7b left; No-GAL4 control starved at 22 °C, 2.4 (1.7–3.6) s, $n = 40$; No-GAL4 control refed at 22 °C, 5.5 (3.9–9.0) s, $n = 53$; IPCs silencing starved at 22 °C, 2.3 (1.6–3.0) s, $n = 44$; IPCs silencing refed at 22 °C, 7.2 (3.9–12) s, $n = 43$). In contrast, the restrictive temperature for the shi^ts-expressing IPCs impaired the refeeding-induced suppression of rolling in the test group, whereas that in the control larvae was unaffected (Fig. 7b right; No-GAL4 control starved at 31 °C, 3.3 (2.5–4.4) s, $n = 49$; No-GAL4 control refed at 31 °C, 7.1 (4.3–14) s, $n = 53$; IPCs silencing starved at 31 °C, 2.2 (1.7–3.0) s, $n = 51$; IPCs silencing refed at 31 °C, 2.2 (1.6–3.3) s, $n = 50$). These results together indicate that Ilps secretion from IPCs triggered by CN neurons in response to ingested sugars is the key step to relay the internal state information to the nociception suppression.

We next investigated the site of action for the sugar-induced Ilps. One potential scenario is that SDGs directly receive the secreted Ilps from IPCs. Since InR is the sole receptor involved in Drosophila insulin signaling[60], we asked whether genetic manipulations of InR specifically in SDGs perturb the refeeding-induced nociceptive suppression. We found that expression of a dominant negative form of InR (InR^DN) in SDGs fully suppressed the (Fig. 7c; No-GAL4 control, Fed, 2.5 (1.9–3.9) s, $n = 48$; Starved, 2.4 (1.5–4.1) s, $n = 57$; Refed, 6.4 (3.6–11) s, $n = 53$; InR^DN expression in SDGs, Fed, 3.8 (2.3–6.4) s, $n = 33$; Starved, 3.7 (2.3–7.1) s, $n = 38$; Refed, 3.0 (2.0–6.9) s, $n = 33$) and the reduction in rolling probability upon mechanical stimulation (Supplementary Fig. 5a), both after sugar refeeding. Conversely, expression of a constitutively active variant of InR (InR^CA) in SDGs increased rolling latency even under normal food conditions (Fig. 7d; No-GAL4 control, 3.0 (2.0–4.4) s, $n = 71$; InR^CA expression in SDGs, 8.6 (5.5–17) s, $n = 68$). To avoid potential effects on developmental processes, we next manipulated InR activity for a limited amount of time by using the temperature sensitive variant of GAL80 (a repressor of GAL4) ubiquitously expressed under the tubulin promoter (tub-GAL80^ts). Since GAL80 cannot bind to split GAL4s, we turned to R16C06-GAL4, the original line found in our screen that cover the neuronal subpopulation including SDGs (Supplementary Fig. 1). For temporal InR inactivation via tub-GAL80^ts, R16C06-GAL4-driven InR^DN expression was suppressed by rearing larvae at low temperature (21 °C). Temperature was then

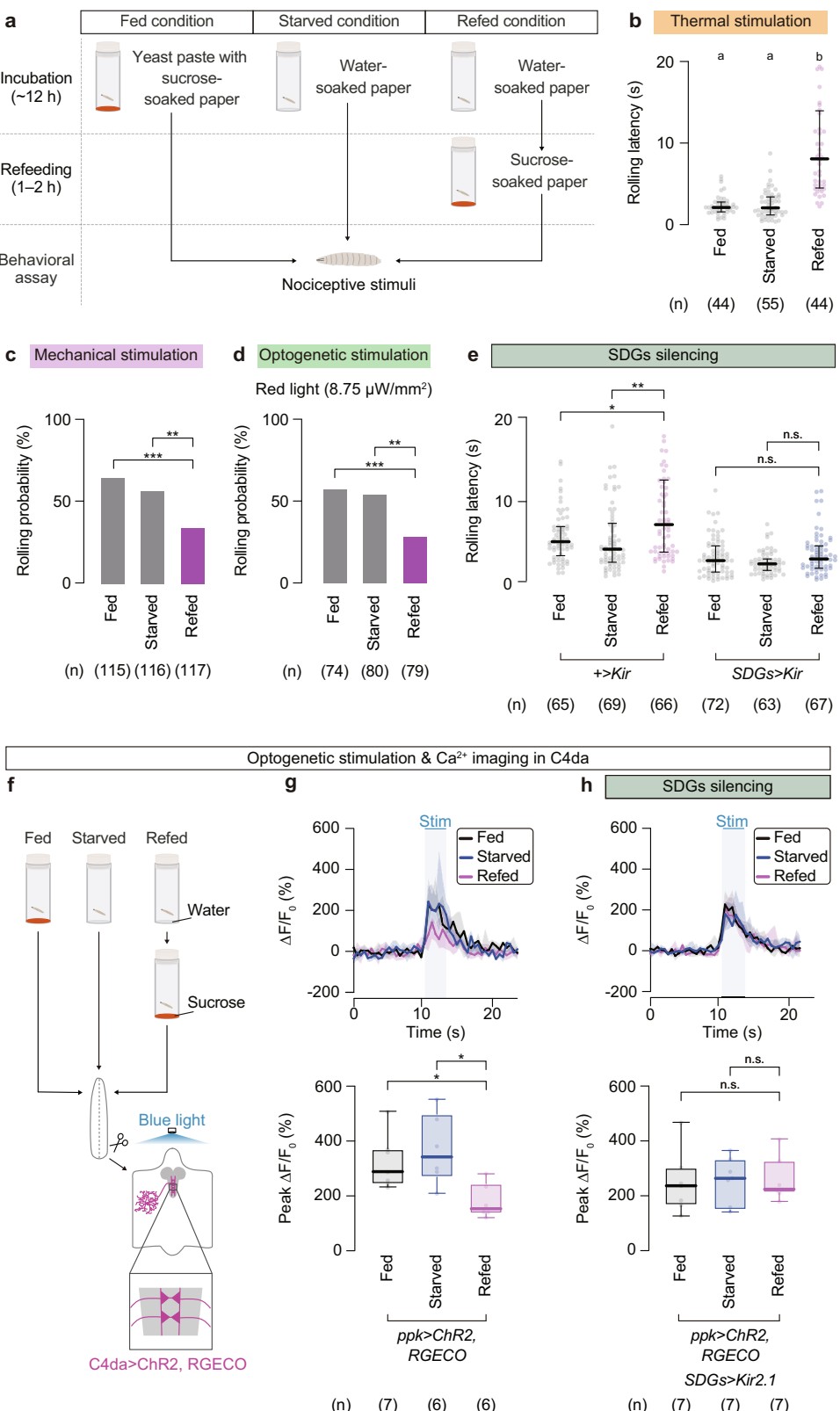

shifted to 30 °C for GAL80$^{ts}$ inactivation during the 6-h starvation followed by 1 h of sugar refeeding. As a result, the refed test group showed a shortened latency of thermo-nociceptive rolling compared to the refed no-*GAL4* control (Supplementary Fig. 5b). This effect was temperature dependent, as both groups exhibited comparable levels of refeeding-induced extension in rolling latency when larvae were kept constantly at 21 °C (Supplementary Fig. 5b). Similarly for

conditional InR activation, larvae harboring *UAS-InR$^{CA}$* with *tub-GAL80$^{ts}$* were reared at 21 °C from the day of egg laying to suppress the *R16C06-GAL4*-driven expression through most of the developmental period. Transiently exposing larvae to 30 °C caused the delay in thermo-nociceptive rolling onset, whereas rearing them at 21 °C without the temperature shift did not (Supplementary Fig. 5c). Collectively, both chronic and time-restricted manipulations of InR activity in SDGs

**Fig. 5 | Sugar feeding is prioritized over escape behavior in starved larvae through SDGs-mediated presynaptic inhibition of C4da neurons. a** Schematic of the feeding experimental setup. **b** Thermo-nociceptive responses of larvae after normal feeding, starvation, or starvation followed by refeeding with sucrose. In this and following panels, 'n' indicates the number of biologically independent animals used for each group, and the thick line and the thin error bar represent the median and the interquartile range, respectively. a, b $p < 0.0001$ (Kruskal–Wallis one-way ANOVA). **c** Mechano-nociceptive rolling probability of larvae under different nutritive conditions. ***$p < 0.0005$, **$p < 0.005$ (Fisher's exact test with Bonferroni correction). **d** Optogenetically induced rolling in larvae under different nutritive states. Channelrhodopsin-expressing C4da neurons were activated by a red-light LED at 8.75 μW/mm$^2$. ***$p < 0.0005$, **$p < 0.005$ (Fisher's exact test with Bonferroni correction). **e** Rolling latency of larvae in which SDGs were genetically silenced

under altered nutritional conditions. **$p < 0.005$, *$p < 0.05$, n.s. $p \geq 0.05$ (Mann–Whitney U-test with Bonferroni correction). **f** Neuronal activity of photo-stimulated C4da neurons under altered feeding conditions, tested either in the wild-type or SDGs-silenced background. The same procedure shown in (**a**) was applied for larval feeding, prior to the ex vivo sample preparation. **g, h** Top, traces of $\Delta F/F_0$. Black, fed; blue, starved; magenta, refed. Note that the blue-shaded box indicates the 3-s optogenetic stimulation time window, starting from 10.5 s and ending at 13.5 s. Bottom, peak $\Delta F/F_0$ changes during the optogenetic stimulation. Box plots indicate the median with the interquartile range, and whiskers represent the minimum-to-maximum range. *$p < 0.05$, n.s. $p \geq 0.05$ (Mann–Whitney U-test with Bonferroni correction). See Supplementary Table 1 for full genotypes. Source data are provided as a Source Data file.

affected the behavioral phenotypes, supporting the notion that InR plays a key role in SDGs mediating the sugar-induced escape modulation.

These behavioral consequences of InR manipulations led us to conduct Ca$^{2+}$ imaging in SDGs, asking whether insulin signaling directly affects their activity levels. In agreement with the behavioral phenotypes, expression of *InR$^{DN}$* lowered the Ca$^{2+}$ levels of SDGs in sugar-refed larvae (Fig. 7e; No-*GAL4* control, Fed, 111 (95.3–130)%, $n = 7$; Refed, 155 (132–184)%, $n = 7$; *InR$^{DN}$* expression in SDGs, Fed, 117 (63.9–134)%, $n = 7$; Refed, 86.5 (72.8–97.5)%, $n = 7$), whereas *InR$^{CA}$*-expressing SDGs exhibited a heightened level of basal activity compared to the genetic control, both fed normally without starvation (Fig. 7f; No-*GAL4* control, 55.6 (50.7–62.6)%, $n = 4$; *InR$^{CA}$* expression in SDGs, 67.9 (62.1–87.0)%, $n = 6$). These data consistently suggest that insulin signaling induces sustained activation of SDGs, leading to nociceptive suppression. Taken all together, these data propose a model in which (i) the nutritional homeostatic system signals through the insulin/InR axis to potentiate the neuronal responsiveness of descending SDGs, and subsequently (ii) tipping the balance towards escape termination via GABA/GABA$_B$-Rs-mediated presynaptic inhibition of peripheral C4da sensory neurons (Fig. 8a, b).

## Discussion

In this study, we identified SDGs, a cluster of descending GABAergic neurons in the larval brain, that negatively regulate noxious stimulus-evoked escape behavior in a context-dependent manner. Upon noxious stimuli, SDGs are transiently activated and provide negative feedback inputs to C4da nociceptive neurons to terminate escape behavior and propel behavioral transition. Furthermore, sugar feeding to starved larvae leads to sustained SDGs activation through the glucose-sensing CN neurons and subsequent insulin signaling, which presynaptically attenuates C4da activity and thereby prioritizes feeding over escape.

### GABAergic descending neurons in the larval brain gate nociception through GABA$_B$ receptor-mediated presynaptic inhibition

Appropriate implementation of gain control in the first synapse plays an important role in sensory processing. In this study, we identified GABAergic SDGs in the larval brain and showed that SDGs play a critical role in gain control of nociceptive inputs at the presynapses on C4da axons. This notion is supported by the following lines of evidence. First, SDGs project descending axons and form synaptic contacts with C4da axonal terminals in the VNC (Fig. 1). Second, immunochemical studies revealed that SDGs are GABA- and Gad1-positive populations, indicating that SDGs are GABAergic neurons. Indeed, *Gad1* RNAi in SDGs dampened their inhibitory modulation on C4da neurons (Fig. 3). Third, GABA$_A$ and GABA$_B$ receptors are expressed in C4da neurons, and specific blockade of GABA$_B$-Rs, but not GABA$_A$-Rs, in C4da neurons enhances nociceptive responses to multiple noxious stimuli (Fig. 3). Fourth, Ca$^{2+}$ imaging revealed that GABAergic inputs attenuate C4da

activation-evoked Ca$^{2+}$ elevation in C4da presynapses (Fig. 4). Lastly, optogenetic activation of SDGs immediately terminates larval rolling behavior (Fig. 2).

Recent studies in rodents identified multiple populations of descending GABAergic neurons in the brainstem that potentially function in pain control. A subpopulation of dual GABAergic and enkephalinergic (GABA$^+$, Penk$^+$) neurons project axons onto sensory afferent terminals in the dorsal spinal cord and suppress behavioral sensitivity to both heat and mechanical stimuli[61]. In contrast, GABAergic but not enkephalinergic (GABA$^+$, Penk$^-$) neurons facilitate mechanical pain by inhibiting local inhibitory neurons in the spinal cord that presynaptically inhibit mechanosensitive primary afferent neurons[62]. Considering the functional and structural characteristics, SDGs in *Drosophila* might play a role analogous to GABA$^+$, Penk$^+$ neurons in rodents. It is also known that GABA$_B$ receptors play a critical role in pain control at the spinal cord, as mice lacking GABA$_B$ receptors exhibit strong hyperalgesia[63] and a spinally administered GABA$_B$ antagonist significantly increases pain responses[64]. Although previous studies suggested multiple roles including presynaptic gating for GABA$_B$ receptors in nociceptive circuits[65,66], underlying mechanisms are not fully understood. Given that SDGs-mediated presynaptic gating in C4da neurons requires GABA$_B$ receptors, further studies in the SDGs-C4da circuit might lead to a better understanding of how GABA$_B$ receptors mediate gain control in nociceptive neurons.

In addition to GABAergic modulations, nociceptive neurons in mammals receive multiple descending modulatory inputs in the spinal cord including serotonergic and norepinephrinergic inputs from the rostral ventromedial medulla and the locus coeruleus, respectively, which positively and negatively fine-tune nociception[67,68]. How these modulatory inputs are orchestrated to regulate nociception in mammals remains elusive, partially due to the complicated arrangement of the mammalian nociceptive circuit in the spinal cord. Similar to the mammalian nociceptive circuit, the *Drosophila* larval nociceptive circuit likely receives multiple types of presynaptic modulations at the first synapses, and each modulation seems to regulate nociception in a context-dependent manner. For example, serotonergic inputs on C4da presynapses contribute to experience-dependent reduction of nociceptive sensitivity in developing larvae[34], whereas neuropeptidergic inputs on C4da presynapses enhance nociceptive responses in mature larvae[35]. The relatively simple structure of the *Drosophila* nociceptive circuit might provide an ideal system to elucidate how multiple modulatory inputs function coordinately as well as independently in nociceptive processing.

### SDGs-mediated nociceptive gating sculps nociceptive escape behavior

In the stereotyped escape behavior, *Drosophila* larvae quickly switch their movement from rolling to fast clawing within seconds[26,28,29]. Strikingly, SDGs silencing as well as blockage of GABAergic inputs from SDGs to C4da neurons significantly prolonged the rolling duration compared to control (Fig. 2), indicating that SDGs activity tunes the

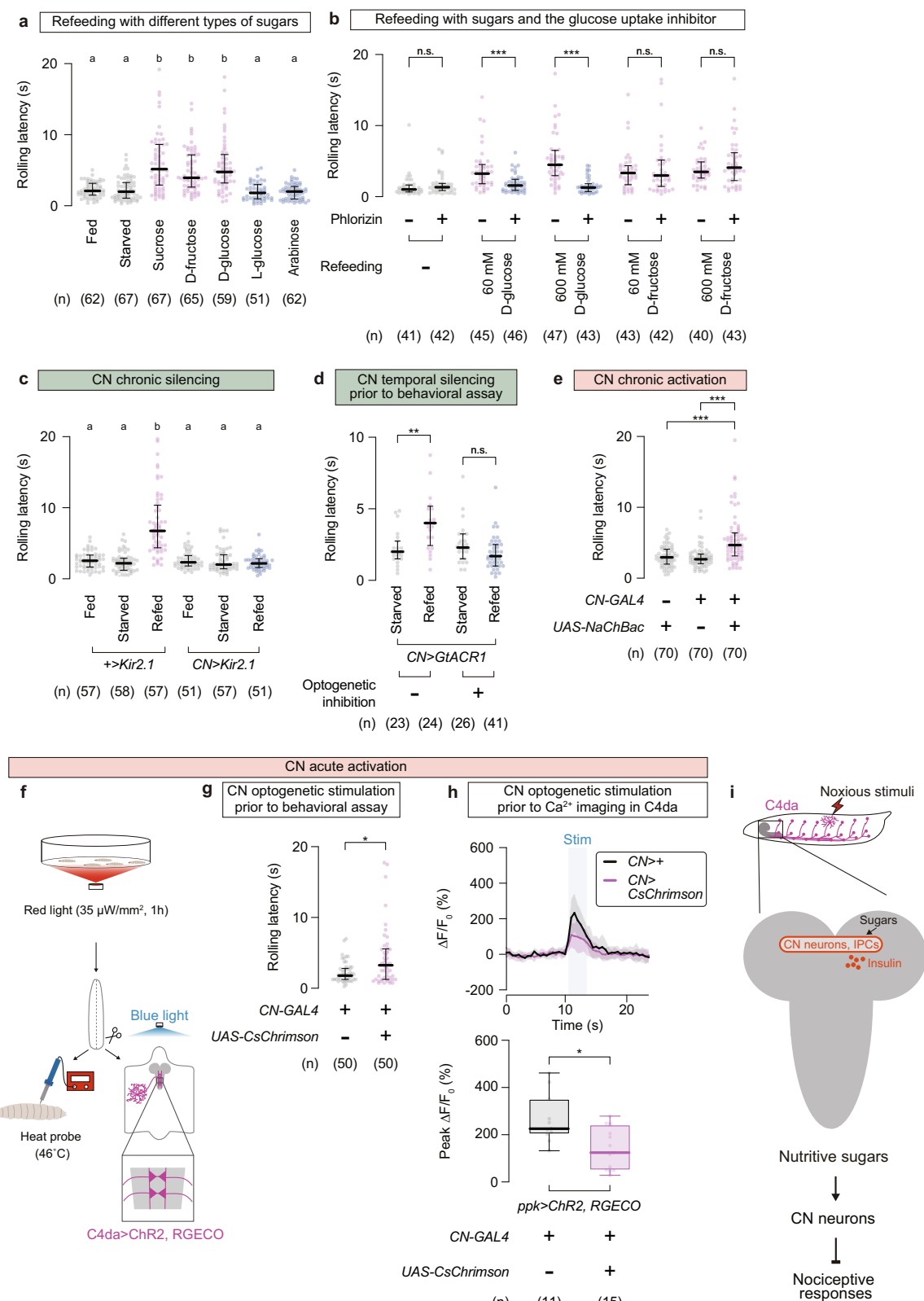

duration of rolling behavior and thus controls orderly transition of escape behaviors. Ca²⁺ imaging of SDGs revealed a transient Ca²⁺ elevation in SDGs axon terminals in response to C4da activation (Fig. 4). It is thus likely that, in response to noxious stimuli, SDGs are activated by C4da-derived nociceptive signals and then provide negative feedback signals to block C4da synaptic transmission (Fig. 8a). This feedback inhibition should contribute to sculp larval escape behavior by

terminating escape rolling immediately after noxious stimuli and thus enabling larvae to propel behavioral transition from rolling to fast crawling at the appropriate moment (Fig. 8a). An important element of this feedback inhibition might be presynaptic GABA$_B$ receptors in C4da neurons. GABA$_B$ metabotropic receptors are major sites for slow synaptic inhibition in the nervous system, providing an extended time window before suppressing synaptic activity compared to GABA$_A$

**Fig. 6 | CN neurons mediate refeeding-induced nociceptive suppression.**
**a** Rolling latency in larvae refed with various types of sugars. Each sugar was refed to starved larvae at the same concentration (600 mM). In this and following panels, 'n' indicates the number of biologically independent animals used for each group, and the thick line and the thin error bar represent the median and the interquartile range, respectively. a, b $p < 0.0001$ (Kruskal–Wallis one-way ANOVA and Dunn's multiple comparison test). **b** Suppressive effect of phlorizin, the glucose uptake inhibitor, on sugar-dependent rolling modulation. Fasted larvae were refed with either D-glucose or D-fructose, each at two concentrations (60 mM and 600 mM). ***$p < 0.0005$, n.s. $p \geq 0.05$ (Mann–Whitney U-test with Bonferroni correction). **c** Thermo-nociceptive responses of larvae in which CN neurons were chronically silenced by Kir2.1 expression. a, b $p < 0.0001$ (Kruskal–Wallis one-way ANOVA and Dunn's multiple comparison test). **d** Effect of CN neurons temporal silencing on the sugar-induced rolling delay. GtACR1-expressing CN neurons were optogenetically inhibited by green light illumination during the 1-h refeeding period, followed by

thermo-nociception assay. **$p < 0.005$, n.s. $p \geq 0.05$ (Mann–Whitney U-test with Bonferroni correction). **e** Thermo-nociceptive responses of larvae in which CN neurons were chronically activated by NachBac expression. ***$p < 0.0005$ (Mann–Whitney U-test with Bonferroni correction). **f–h** Behavioral and physiological effects of CN neurons temporal activation. Starved larvae expressing CsChrimson in CN neurons were first illuminated with red light for 1 h without refeeding (**f**). Subsequently, larvae were subjected to either thermo-nociception assay (**g**) or Ca$^{2+}$ imaging in C4da neurons coupled with optogenetic stimulation (**h**). Black, CN>+; magenta, CN>CsChrimson. Note that CN and C4da neurons expressed different channelrhodopsins (red-light responsive CsChrimson and blue-light responsive ChR2) enabling independent stimulation. Box plots show the median with an interquartile range, and whiskers represent the minimum-to-maximum range. *$p < 0.05$ (Mann–Whitney U-test). See Supplementary Table 1 for full genotypes. Source data are provided as a Source Data file. **i** A potential pathway for sugar refeeding-mediated suppression of nociceptive responses.

ionotropic receptors[65]. Although the timescale of C4da synapse activity and the dynamics of GABA release from SDGs require further investigation, GABA$_B$ receptors might ensure the translation of transient neural signals into more sustained cellular and behavioral responses across time scales.

In addition to the behavioral transition, SDGs control nociceptive sensitivity, as silencing SDGs sensitized larvae to multiple noxious stimuli (Fig. 2). In the olfactory system, presynaptic gating by GABAergic neurons defines the dynamic range of the olfactory sensitivity and fine-tunes olfactory behavior[48,52]. Similarly, fine-tuning the gain for noxious cues should be important for adjusting an organism's sensitivity to its environments. To maximize chances of survival, animals need to flexibly decide behavioral priorities, especially in natural environments. This form of nociceptive gain control should contribute to a flexible decision making between competitive behaviors such as escape from danger and searching for energy sources.

**Glucose-sensing neurons promote sustained SDGs activation through insulin signaling**
In this study, we found that sugar refeeding consistently and dramatically attenuates escape behavior to thermal and mechanical insults in larvae (Fig. 5). By contrast, hunger alone showed no significant effect on larval escape behavior (Fig. 5). Similarly, refeeding in mice (2-h feeding after 24-h food deprivation) suppressed both acute and chronic pain[69], whereas hunger (24-h food deprivation) had less effect on acute pain by thermal, mechanical, and chemical insult while selectively abolishing inflammatory pain responses[70]. It is thus likely that refeeding and hunger have distinct impacts on nociceptive responses in both flies and mammals.

Notably, nutritive sugars such as D-glucose and D-fructose effectively suppressed escape behavior, whereas nonnutritive sugars, L-glucose and arabinose, had no significant analgesic effect in insulted larvae (Fig. 6). This nutritive sugar-specific nociceptive suppression is mediated, at least in part, by the glucose-sensing CN neurons, as CN neurons are selectively activated by nutritive sugars such as D-glucose and D-fructose and inhibited by the glucose transporter antagonist phlorizin[36]. Indeed, both silencing CN neurons by Kir2.1 and phlorizin application significantly attenuated feeding-induced nociceptive suppression (Fig. 6). Given that CN neurons directly innervate IPCs and induce Ilps secretion from IPCs in response to acute elevation of D-glucose levels in hemolymph[36], it is likely that sugar feeding engages SDGs in nociceptive gating through activation of the sugar-sensing CN neurons in response to acute D-glucose elevation in the brain and subsequent insulin signaling derived from IPCs (Fig. 6). Consistent with this notion, expression of a dominant negative InR in SDGs blocked refeeding-evoked SDGs activity and thus feeding-induced nociceptive suppression, whereas expression of a constitutively active InR upregulated SDGs activity and induced nociceptive suppression even in non-feeding condition (Fig. 7).

Recent studies revealed that insulin and insulin-like growth factor (IGF) have a wide variety of brain functions in vertebrates and invertebrates[71]. In *Drosophila*, insulin signaling transcriptionally regulates expression levels of sNPF receptors and Tachykinin receptors in particular classes of the olfactory receptor neurons, which facilitates food-searching behavior[72,73]. Similarly, insulin signaling regulates expression levels of multiple neuropeptides in mammalian hypothalamic neurons[74,75]. Recent studies suggest that multiple different types of glucose-sensing neurons exist in the mammalian brain regions including the hypothalamus, amygdala, and brainstem[76], although their physiological functions in vivo remain largely unknown. It is thus of great interest to investigate whether insulin/IGF signaling and the glucose-sensing neurons might be involved in pain suppression through descending neurons in the mammalian brain.

## Methods
### Fly Strains
The following strains of *Drosophila melanogaster* were obtained from Bloomington *Drosophila* Stock Center: *w$^{1118}$* (BL#3605), *UAS-CD4-tdTomato* in VK00033, *UAS-Stinger* (BL#90914), *ppk-CD4-tdTomato* on 2nd chromosome (BL#35844), *ppk-CD4-tdTomato* on 3rd chromosome (BL#35845), *10XUAS-mCD8::GFP* in attP2 (BL#32184), *20XUAS-IVS-mCD8::GFP* in attP2 (BL#32194), *20XUAS-CsChrimson::mCherry* in su(Hw)attP5 (BL#82181), *UAS-nsyb-spGFP$_{1-10}$*, *LexAop-CD4-spGFP$_{11}$* (BL#64314), *R16C06-GAL4* in attP2 (BL#48719), *R21F01-p65.AD* in attP40, *R93B07-GAL4.DBD* in attP2 (BL#69254), *10XUAS-mCD8::RFP* in attP18; *13XLexAop2-mCD8::GFP* in su(Hw)attP8 (BL#32229), *UAS-Kir2.1::EGFP* (BL#6596), *LexAop-ReaChR* in VK00005 (BL#53746), *UAS-GtACR1::EYFP* in attP2 (BL#92983), *UAS-NaChBac::EGFP* (BL#9466), *13XLexAop-IVS-jGCaMP7s* in VK00005 (BL#80913), *Gad1-Trojan-LexA-QFAD* (BL#60324), *CaryP* in attP40 (BL#36304), *UAS-IR-Gad1* (*TRiP.HMC03350*) in attP40 (BL#51794), *UAS-IR-Gad1* (*TRiP.JF02916*) in attP2 (BL#28079), *UAS-Dcr-2* on 2nd (BL#24650) or 3rd (BL#24650) chromosome, *GABA$_B$-R1$^{2A-GAL4}$* (BL#84701), *GABA$_B$-R2$^{2A-GAL4}$* (BL#84634), *CaryP* in attP2 (BL#36303), *UAS-IR-GABA$_B$-R1* (*TRiP.HMC03388*) in attP2 (BL#51817), *UAS-IR-GABA$_B$-R2* (*TRiP.HMC02975*) in attP2 (BL#50608), *GABA$_B$-R3$^{2A-AD-GAL4}$* (BL#84635), *Rdl$^{2A-GAL4}$* (BL#84688), *UAS-IR-GABA$_B$-R3* (*TRiP.HMC02989*) in attP40 (BL#50622), *UAS-IR-Rdl* (*TRiP.HMC03643*) in attP40 (BL#52903), *13XLexAop2-IVS-NES-jRGECO1a-p10* in su(Hw)attP5 (BL#64426), *13XLexAop2-ChR2.T159C-HA* in VK00013 (BL#52256), *20XUAS-ChR2.T159C-HA* in VK00018 (BL#52258), *ilp2-GAL4* (BL#37516), *UAS-InR.K1409A* as InR$^{DN}$ (BL#8252), *UAS-InR.Del* as InR$^{CA}$ (BL#8248), *tub-GAL80$^{ts}$* (BL#7017), *20XUAS-IVS-jGCaMP7s* in su(Hw)attP5 (BL#80905), *20XUAS-IVS-CsChrimson::mCherry* in VK00005 (BL#82180), and the CaLexA reporter (*LexAop-CD8-GFP-2A-CD8-GFP; UAS-mLexA-VP16-NFAT, LexAop-rCD2-GFP*) (BL#66542). The following RNAi lines were obtained from Vienna *Drosophila* Resource Center: GD control (VDRC#60000), KK control (VDRC#60100), *UAS-IR-Gad1* (*GD8508* on 2nd chromosome, VDRC#32344), *UAS-IR-GABA$_B$-R1*

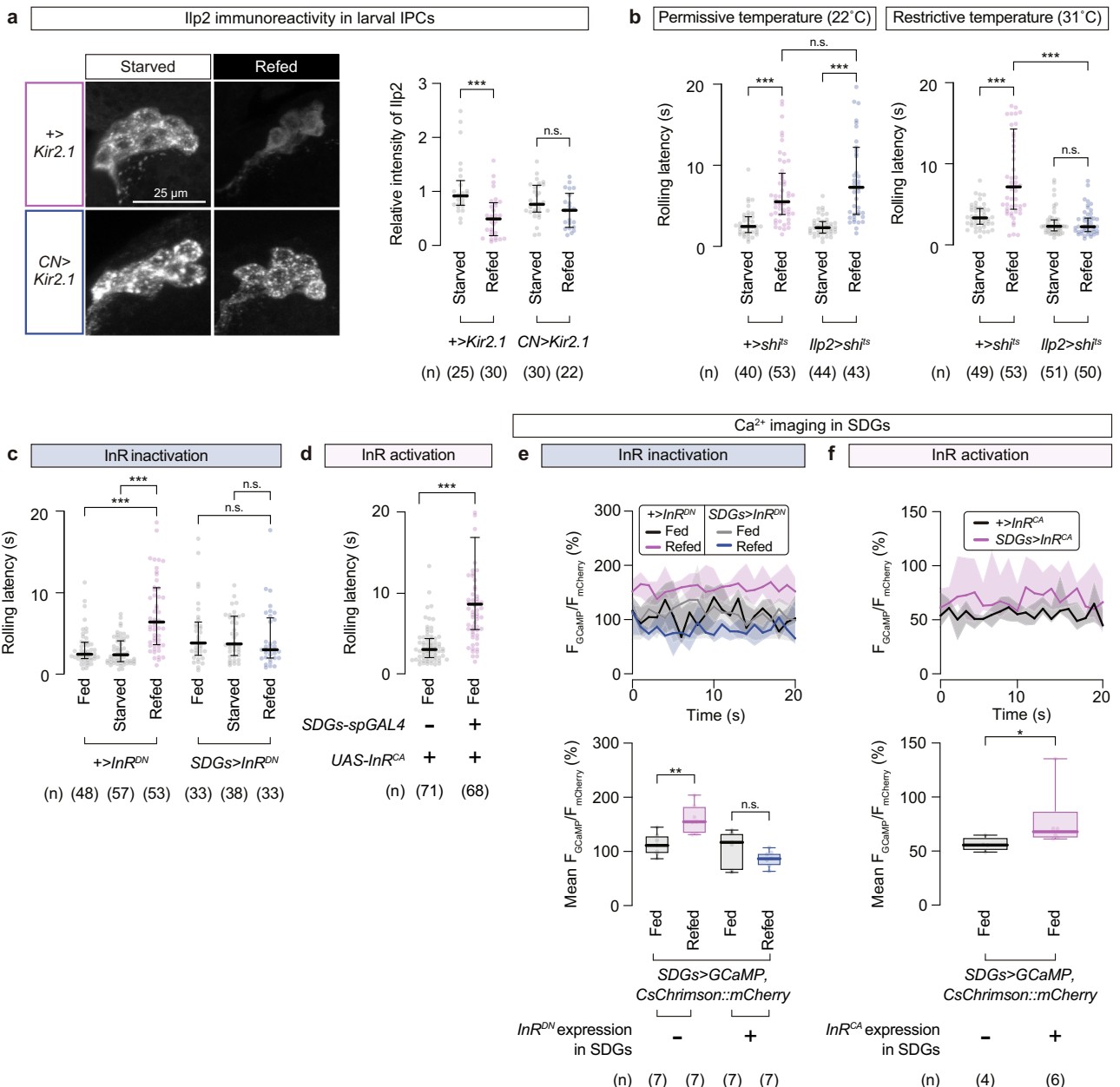

**Fig. 7 | Insulin signaling in SDGs enhances SDG activity and suppresses escape behavior. a** CN neurons-dependent alteration of Ilp2 immunoreactivity in IPCs after sugar refeeding. Left, representative images of larval IPCs stained with the anti-Ilp2 antibody. Right, relative intensities of Ilp2 signals measured within IPCs. In this and following panels, 'n' indicates the number of biologically independent animals used for each group, and the thick line and the thin error bar represent the median and the interquartile range, respectively. *** $p < 0.0005$, n.s. $p \geq 0.05$ (Mann–Whitney U-test with Bonferroni correction). **b** Refeeding-dependent modulation of thermo-nociceptive responses in larvae with temporal silencing of IPCs. Larvae were either starved or refed at two different temperatures; under permissive (left, 22 °C) or restrictive (right, 31 °C) conditions for *shibire*$^{ts}$. *** $p < 0.0005$, n.s. $p \geq 0.05$ (Mann–Whitney U-test with Bonferroni correction). **c** Thermo-nociceptive responses in larvae with InR function impaired specifically in SDGs. InR$^{DN}$, the dominant negative form of InR. *** $p < 0.0005$, n.s. $p \geq 0.05$ (Mann–Whitney U-test with Bonferroni correction). **d** Rolling latency of normally fed larvae with artificially

enhanced InR activity in SDGs. InR$^{CA}$, the constitutively activated variant of InR. *** $p < 0.0005$ (Mann–Whitney U-test). **e** Basal neuronal activity of SDGs expressing the dominant negative InR. Samples were prepared from larvae fed with normal food. Top, traces of $F_{GCaMP}/F_{mCherry}$, calculated by normalizing the signal intensity of GCaMP with that of CsChrimson::mCherry expressed as a marker. Thick lines and accompanying shades represent the median and the interquartile range, respectively. Black, +>InR$^{DN}$ fed; magenta, +>InR$^{DN}$ refed; gray, SDGs > InR$^{DN}$ fed; blue, SDGs > InR$^{DN}$ refed. Bottom, mean $F_{GCaMP}/F_{mCherry}$ values from the 10-s recording windows. Box plots show the median with an interquartile range, and whiskers represent the minimum-to-maximum range. ** $p < 0.005$, n.s. $p \geq 0.05$ (Mann–Whitney U-test with Bonferroni correction). **f** SDGs activity with enhanced InR functions. Black, +>InR$^{CA}$; magenta, SDGs > InR$^{CA}$. * $p < 0.05$ (Mann–Whitney U-test). See Supplementary Table 1 for full genotypes. Source data are provided as a Source Data file.

(KK109166) in VIE260b (VDRC#101440), *UAS-IR-GABA$_B$-R2* (KK100020) in VIE260b (VDRC#110268). *ppk-nlsLexA::p65* in attP2[28], *ppk-GAL4* on X chromosome[77], and *20XUAS-droRGECO*[33] were generated in our lab; *UAS-brpD3::mCherry* was from T. Suzuki; *UAS-PTX* was from G. Roman[47];

*CN-GAL4* (VT58471-GAL4, Cha-GAL80) was from G. S. B. Suh[36]; *UAS-shibire*$^{ts}$ was from T. Kitamoto[78]. *R16C06-LexA* was created by cloning the enhancer region of *R16C06-GAL4* into the pBPLexA::p65Uw vector (Addgene #26231), followed by phiC31-mediated transgenesis targeting

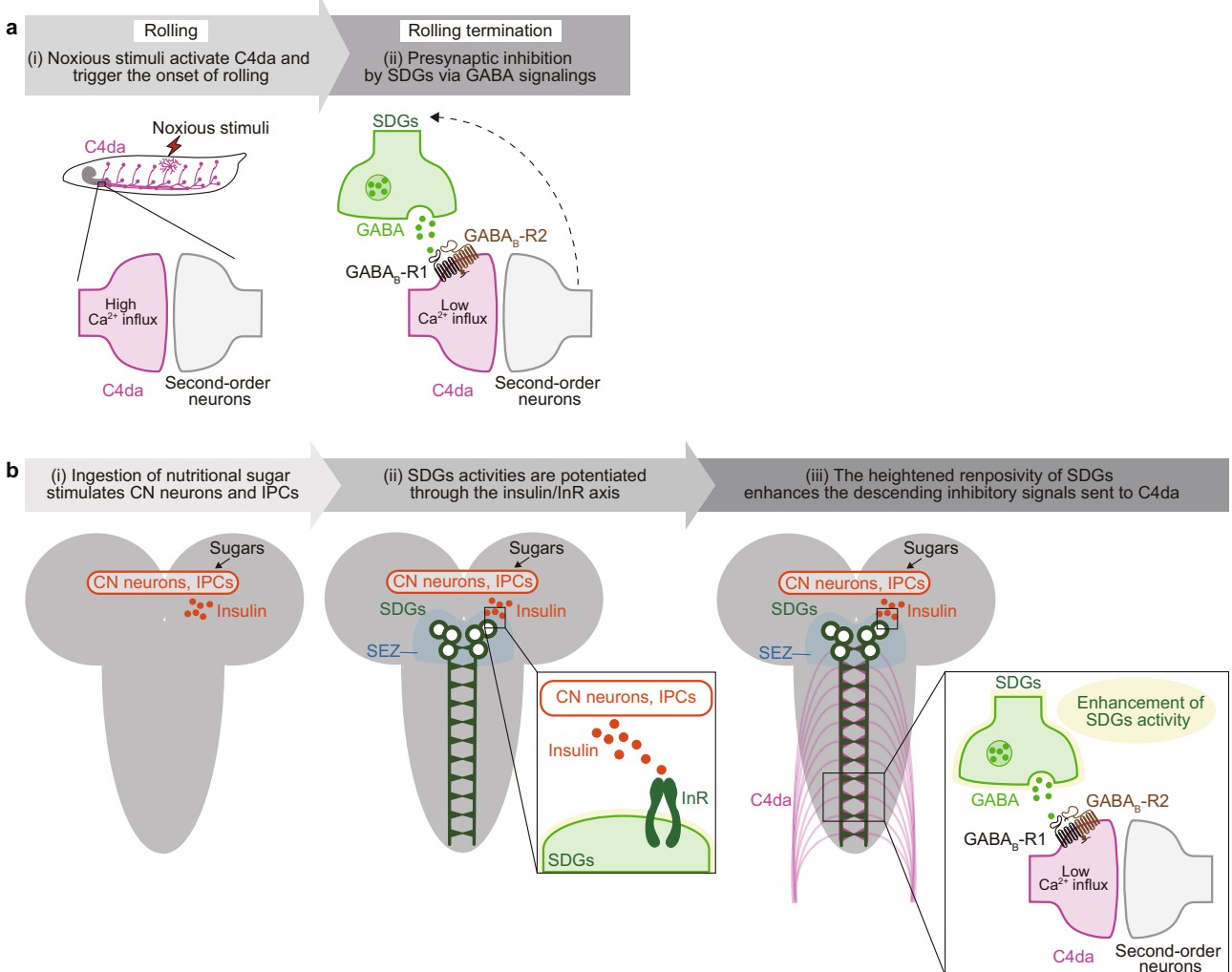

**Fig. 8 | A schematic model of presynaptic inhibition of C4da and sugar signaling in SDGs. a** Presynaptic inhibition by SDGs via GABA signaling. Noxious stimuli first activate the peripheral C4da neurons, followed by subsequent activation of downstream secondary neurons that trigger the onset of larval rolling. During this process of nociceptive information relay, SDGs become activated and in turn send GABAergic inhibitory inputs to C4da neurons via GABA$_B$-Rs. This SDGs-mediated negative feedback induces rolling termination with a time lag, which enables behavior transition for efficient escape from danger. **b** SDGs mediate refeeding-dependent nociceptive suppression via insulin/InR signaling. Ingestion of nutritional sugar stimulates the metabolic neuronal system in the central brain, comprising the glucose-sensing CN neurons and insulin-producing cells (IPCs). Neuronal activities in SDGs, of which cell bodies are located at the SEZ brain region, are subsequently potentiated through the insulin/InR axis. The heightened responsiveness of SDGs enhances the descending inhibitory signals sent to C4da neurons, tipping the balance towards escape termination and feeding prioritization.

the attP40 insertion site (BestGene Inc.). Animal experiments using transgenic flies were conducted with the approval of Research Ethics Committee of the University of Tokyo.

## Immunohistochemistry
Third-instar larvae were dissected in phosphate buffered saline (PBS), and brains were fixed in 4% paraformaldehyde/PBS for 30 min at room temperature. For direct observation of fluorescence-labeled neurons, brains were washed in 0.1% Triton X-100/PBS (PBST) for 1 h at room temperature after fixation and then submerged in VECTASHIELD mounting medium (H-1000, Vector Laboratories) for at least 10 min. For immunohistochemical analyses, larval brains were washed for 5 times with PBST and were incubated in 5% normal goat serum (NGS)/PBST or 5% bovine serum albumin (BSA)/PBST (for Ilp2 detection) at room temperature for 2 h. Samples were subsequently incubated at 4 °C overnight with primary antibodies diluted in 5% NGS/PBST or 5% BSA/PBST (for Ilp2 detection). Followed by PBST washes for 5 times, samples were further incubated at 4 °C overnight with secondary antibodies diluted in 5% NGS/PBST or 5% BSA/PBST (for Ilp2 detection).

Brains were then washed and mounted as above. Fluorescence images were obtained by confocal microscope (TCS SP8, Leica). The following primary antibodies were used: mouse anti-GFP (1:200, clone 3E6, A11120, Thermo Fisher Scientific), mouse anti-ChAT (1:50, 4B1, Developmental Studies Hybridoma Bank), rabbit anti-GABA (1:100, A2052, Sigma-Aldrich), rabbit anti-VGluT (1:400; a gift from H. Aberle[79]), rabbit anti-Ilp2 (1:2000; a gift from T. Nishimura[80]). Fluorescently conjugated secondary antibodies used were goat anti-mouse IgG (H + L) Alexa Fluor 635 (1:500, A31574, Thermo Fisher Scientific), and goat anti-rabbit IgG (H + L) Alexa Fluor 633 (1:500, A21071, Thermo Fisher Scientific).

## Behavioral assays
Experiments were performed in a blinded fashion where the experimenter was unaware of larval genotypes. All movies were manually checked frame-by-frame. ImageJ was used to load the movies, saved as avi or multi-tiff files, to observe each frame without any special processing. Manual detection of the rolling occurrence, defined as a "corkscrew-like" behavior in which a larva turned on its long axis more

than 90°, was followed by quantification of rolling parameters. Duration and latency were measured based on time stamps extracted from recorded movies using the VLC media player (available at https://www.videolan.org/vlc/index.en_GB.html) with an extension plugin "Time v3.2" (available at https://addons.videolan.org/p/1154032/).

**Mechanical stimulation.** Stage- and density-controlled 3rd instar larvae (72 ± 2 h after egg laying) were prepared by collecting eggs from 20 females (mated with 5 males for at least 2 d, kept under a 9AM:9PM light:dark cycle) within a 6-h time window starting from afternoon. Twelve milliliter of 2% agarose (313-90231, Nippon gene) was solidified in a plastic dish with a diameter of 9 cm, and 2 mL of water was poured before the assay to make the surface wet. Larvae were gently washed with water and placed on the agarose plate. Forward-locomoting larvae were stimulated at mid-abdominal segments (A4–6) with a 45–50 mN von Frey filament (Omniflex monofilament fishing line (6 lb test, diameter 0.009 inches, Shakespeare)) twice within 2 s. Movies were recorded under a microscope (SZX7 with an adapter U-TV0.8XC, Olympus) equipped with a camera (Visual IV PRO2 LITE, Visualix or HDWi-200E, Relyon) at approx. 30 fps. For optogenetic activation of *CsChrimson*-expressing SDGs during mechanical stimulation, ATR-fed larvae were observed under a stereomicroscope (MVX10, Olympus) with infrared background illumination (LDR2-132IR850-LA, CCS) and were stimulated using a 640-nm LED (93.3 μW/mm$^2$, Lumencor Spectra X7, Lumencor). Movies were recorded by a sCMOS-Camera (Zyla 5.5, Andor) at 20 fps.

**Thermal stimulation.** Stage- and density-controlled 3rd instar larvae (96 ± 2 h after egg laying) were prepared as above. Larvae were gently washed with water and placed on the surface-wet agarose plate. The heat probe, set at 46 ± 0.5 °C with a custom-built thermocouple device, was applied to forward-locomoting larvae at mid-abdominal segments (A4–6) until the execution of nociceptive rolling. In case where rolling was not observed for more than 20 s after the probe attachment, the rolling latency was analyzed as 20 s. Movies were recorded under the same conditions as the mechano-nociception assay.

**Optogenetic activation/inhibition.** Larvae were grown in the standard medium containing 1 mM all-*trans*-retinal (ATR; R2500, Sigma-Aldrich) at 25 °C. Third-instar wandering larvae were washed with deionized water. Five to ten larvae were then placed at the center of the arena made of 1% agarose. To stimulate red-shifted channelrhodopsins ReaChR and CsChrimson, larvae were exposed to red light (617 nm, 35 μW/mm$^2$). For optogenetic inhibition using the anion channelrhodopsin GtACR1, green light (575 nm, 30 μW/mm$^2$) was applied. Movies were recorded using a CCD-Camera (1500M-GE, Thorlabs) at 2 fps. Rolling duration following the photo-stimulation was measured from initiation to termination of the first roll.

**Chemical stimulation.** Allyl isothiocyanate (AITC; I0185, Tokyo Chemical Industry) was used as a nociceptive agent. Third-instar wandering larvae were washed with deionized water. Five to ten larvae were placed at the center of a plastic plate with a diameter of 10 cm. The substrate was poured with 500 μl of 500 mM AITC. Red light illumination and movie recordings were conducted under the same conditions as the optogenetic activation experiments.

**Ca$^{2+}$ imaging**
Larvae were reared from eggs on a standard medium containing 1 mM ATR (R2500, Sigma-Aldrich). Third-instar wandering larvae were pinned down on a silicon dish (Silpot 184, Dow Corning Toray) and were dissected along the dorsal midline in either calcium-free HL3.1 buffer[81] (70 mM NaCl, 5 mM KCl, 4 mM MgCl$_2$, 10 mM NaHCO$_3$, 5 mM trehalose, 115 mM sucrose, 5 mM HEPES, pH 7.2), sugar-free buffer[82] (108 mM NaCl, 5 mM KCl, 8.2 mM MgCl$_2$, 4 mM NaHCO$_3$, 1 mM

NaH$_2$PO$_4$, 5 mM HEPES, pH 7.5), or sugar-free buffer with GABA (γ-Aminobutyric acid, A2129, Sigma-Aldrich). Note that using the "calcium-free" buffer was key to avoid brain movements due to larval muscle contractions. Internal organs except for neural tissues were removed. Axon terminals of either C4da or SDGs at the A5–6 segments in the VNC were imaged using a microscope (BX51WI, Olympus) equipped with a spinning-disk confocal unit (Yokogawa CSU10, Yokogawa) and an EM-CCD digital camera (Evolve, Photometrics).

For activation of C4da neurons expressing the light-gated channelrhodopsin ChR2, blue light with a wavelength of 475 nm and a power of 208 μW/mm$^2$ was delivered by the pE-300 device (CoolLED). See Supplementary Table 2 for full imaging conditions. Obtained images were analyzed using Metamorph (ver. 7.10.4.407 and offline ver. 77.5.0, Molecular Devices) and ImageJ (NIH) software. To obtain stable Ca$^{2+}$ signals, each image was manually checked and motion correction was applied using the StackReg/TurboReg plugins[83] (available at http://bigwww.epfl.ch/thevenaz/stackreg/) for ImageJ. Four ROIs, each with a diameter of 1 μm per one segment of either side (depicted in the schema of Fig. 1a as the triangle-shaped region of C4da axon terminal), were set. After the mean signal intensity within each ROI was measured using ImageJ, values from 4 ROIs were averaged to represent the data of one brain sample. The number of brain preps (not the number of ROIs) for each genotype/treatment group was shown as (n) within figures. As a baseline, F$_0$ was calculated by averaging the fluorescent signals for the first 10 s prior to onset of optogenetic stimulation. Calcium transient ΔF/F$_0$ was calculated by normalizing the raw signal intensity F at each timeframe to F$_0$ with the following formula: $\Delta F/F_0 = (F - F_0)/F_0$.

For Ca$^{2+}$ imaging in SDGs, dissection and imaging were performed as above except that ROIs were set on the neurites of SDGs at the A5–6 segments. Signal intensity of the calcium indicator GCaMP (F$_{GCaMP}$) and the neurite-labeling CsChrimson::mCherry (F$_{mCherry}$) were alternately measured. See Supplementary Tables 1 and 2 for full genotypes and detailed imaging conditions, respectively.

**Nutritional conditions**
Third-instar larvae were prepared as described for the thermal nociception assays. For fed or starved conditions, larvae were incubated at 25 °C for ~12 h either on yeast paste plus 600 mM sucrose-soaked papers (200 μl of liquid applied to kimwipes cut to 1 cm × 1 cm) or on water-soaked papers, respectively. For refeeding, starved larvae were transferred to vials containing papers soaked with 200 μl of either sucrose (196-00015, Wako), D-fructose (127-02765, Wako), D-glucose (041-00595, Wako), L-glucose (G0226, Tokyo Chemical Industry), L-arabinose (A3256, Sigma-Aldrich) (each at 600 mM) or phlorizin dihydrate (P3449, Sigma-Aldrich) for 1–2 h. All larvae were washed with water and subjected to nociception assays as described above.

**Larval locomotion activity**
For Supplementary Fig. 2c, locomotion speed was calculated from the 10-s pre-stimulation windows before the optogenetic activation in Fig. 2a. For Supplementary Fig. 2d, each movie was newly recorded for 10 s to measure the locomotion speed. For Supplementary Fig. 4a, larvae were placed on 2% agarose plates without food source after each nutritional treatment described in Fig. 5a. Movies were recorded for 2 min with the same camera used in optogenetic experiments (see "*Optogenetic activation/inhibition*" described above), and locomotion speed was measured from the last 10 s of the recorded data.

**Transcription-based measurement of neuronal activity**
The CaLexA system[58] was applied to measure the activities of CN neurons after sugar refeeding, essentially according to the previous report in adults[36]. Parental *CN-GAL4* and the CaLexA reporter lines were crossed, and offspring larvae were reared to third instar. Collected larvae were starved on a water-soaked paper for 12 h and

subsequently refed in a 600 mM sucrose vial, as described above. Brains were dissected and the native fluorescence of GFP signals were observed under the confocal microscope. Signal intensity at the cell body was measured by ImageJ.

## Temporally-controlled neuronal silencing

Larvae harboring *UAS-shibire^(ts)* were raised at 22 °C for 5 d (instead of 4 d in normal assays raised at 25 °C) after egg collection. For restrictive temperature treatment, larvae were starved on water-soaked papers at 22 °C for 6 h, followed by a pre-warming step at 31 °C for 30 min. Larvae were then quickly transferred onto either water- or sucrose-soaked papers and were subsequently refed at 31 °C for 2 h. The thermo-nociception assay was performed as above.

## Temporally-controlled manipulation of InR activity

Parental *R16C06-GAL4* and either *UAS-InR^(DN)*; *tub-GAL80^(ts)* or *UAS-InR^(CA)*; *tub-GAL80^(ts)* lines were mated for 2 d under the standard condition. Parents were then transferred to fresh food vials for egg collection. To increase the yield of offspring, parents were kept at 25 °C (instead of 21 °C) during 9 h of egg laying. Collected embryos were reared at 21 °C to prevent the *R16C06-GAL4*-driven expression. For temporal InR inactivation, larvae were kept at 21 °C for 5 d (note that it takes additional 1 d to reach the 3rd instar compared to the standard condition at 25 °C) until the last ~7 h preceding behavior assays. Temperature was subsequently shifted to 30 °C, the restrictive temperature for GAL80^(ts), from the beginning of the 6-h starvation and through the end of the 1-h sugar refeeding. For conditional InR activation, larvae were reared at 21 °C for 96 h, until the day before the behavior assays. For the following 15 h, larvae underwent an over-night warming at 30 °C to disrupt the GAL80^(ts) function, thus allowing InR^(CA) expression. After the high-temperature treatments, larvae were briefly washed and immediately tested for thermal nociceptive responses at room temperature.

## Statistical analysis

Statistical analyses by Kruskal–Wallis one-way ANOVA, Mann–Whitney U-test, or Fisher's exact test were performed by Prism 9.4.1 (RRID:SCR_002798, GraphPad). All statistical tests were two-sided. Bonferroni correction was applied for multiple comparisons. Asterisks (*) represent $p$-values as indicated within each figure legend. All data necessary to reproduce the figure panels and statistical analyses are available as Source Data file.

## Reporting summary

Further information on research design is available in the Nature Portfolio Reporting Summary linked to this article.

# Data availability

Raw data reported in this paper will be shared by the lead contact upon request. Any additional information required to reanalyze the data reported in this paper is available from the lead contact upon request. Source data are provided with this paper.

# Code availability

No custom code was used. Publicly available codes are listed below: The StackReg/TurboReg plugins for ImageJ, available at http://bigwww.epfl.ch/thevenaz/stackreg/. The extension plugin "Time v3.2" for VLC, available at https://addons.videolan.org/p/1154032/

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

## Acknowledgements

We thank Y. Aso, J. Truman, and J. Parrish for valuable suggestions; T. Suzuki, G. Roman, G. S. B. Suh, T. Kitamoto, and Bloomington *Drosophila* Stock Center for fly stocks; H. Aberle for the anti-VGluT antibody; T. Nishimura for the anti-Ilp2 antibody; H. Ito, N. Yamaji, M. Maetani, A. Ogasawara, E. Kato, T. Rikiishi, S. Ando, M. Hayashi, and S. Miyazaki for technical assistance; the members of Emoto Lab for critical comments and discussion. This work is supported by MEXT Grants-in-Aid for Scientific Research on Innovative Areas "Dynamic regulation of brain function by Scrap and Build system" (KAKENHI 16H06456), JSPS (KAKENHI 16H02504), WPI-IRCN, AMED-CREST (JP22gm310010), JST-CREST, Toray Foundation, Naito Foundation, Takeda Science Foundation, and Uehara Memorial Foundation to KE; the Leading Initiative for Excellent Young Researchers (LEADER) from MEXT, JSPS (KAKENHI 22K06309), and AMED-PRIME (JP22gm6510011) to K.I.; Grant-in-Aid for JSPS Fellows (20J22063) to M.N.D.

## Author contributions

M.N.D., K.I. and J.Y. performed the experiments and data analyses; M.N.D., K.I. and K.E. are responsible for study conception, experimental design, and data interpretation; M.N.D., K.I. and K.E. wrote the original manuscript; J.Y. and M.T. edited the paper.

## Competing interests

The authors declare no competing interests.
