## [Peer Review File · Nature Communications]

Descending GABAergic pathway links brain sugar-sensing to peripheral nociceptive gating in *Drosophila*REVIEWER COMMENTS

Reviewer #1 (Remarks to the Author):

In this manuscript, Nakamizo-Dojo et al. report a neural mechanism by which feeding of starved *Drosophila* larvae overrides the escape behaviour elicited by noxious stimuli. They first report a cluster of GABAergic neurons in the larval brain that suppress nociceptive inputs through GABAB receptors in the nociceptors. They then found that glucose feeding to starved larvae activates these neurons through glucose-sensing neurons and insulin signaling. This attenuates the escape behaviour in response to noxious stimuli.

The discovery of the SDG descending GABAergic neurons in *Drosophila* is important. Although connectome studies have indicated the existence of descending neurons, physiological and functional investigations have not been done to establish the existence and functional significance of such neurons. The finding the sugar-sensing neurons in the brain inhibits SDG neurons is also novel and important, as it offers mechanistic insights into the suppression of nociception by feeding at the circuit level. While studies in mammals have shown that feeding suppresses nociceptive responses and uncovered the circuitry and molecular mechanisms, this study is the first in *Drosophila*, which is still novel and significant.

Overall, the studies are well designed (though see concerns below) and the paper well written. The use of multiple ways to stimulate the nociceptors strengthens the study and demonstrates the regulation is across nociceptive modalities. However, there are many technical concerns.

Major concerns:

(1) To inhibit neurons, authors expressed Kir2.1 in target neurons throughout development. Because inhibiting neurons during development may change neuronal activity and circuitry, acute inhibition should be used in at least some key experiments. Such approaches are available and used in *Drosophila*, for example, GtACR. Reviewer recognizes that some experiments in this paper may not be possible to use GtACR, but some are. For example, the behaviour experiments.

(2) Evidence showing that Gad1 knockdown reduces Gad1 expression is needed. If a Gad1 antibody isn't available, GABA staining should be used to test Gad1 knockdown. Moreover, one RNAi line is insufficient to establish the function of a gene. Additional RNAi lines, rescue with transgenes, or genetic mutants should be used to corroborate the claimed functions of Gad1 and GABABR1/R2.

(3) Negative controls for GABA and Gal1 staining in Fig 3a are missing.

(4) The evidence that CN neurons mediate refeeding-induced nociceptive suppression is weak. Data showing that CN neuron activity changes in the Refed group is missing. Moreover, if authors' model is correct, then activating CN neurons should inhibit nociceptive behaviour and C4da responses.

(5) The proper control for the refeed experiment should be sucrose-soaked paper only. The addition of yeast paste complicates the experiment, especially considering the report that dendrites of C4da neurons hyperarborize on a low-yeast diet (Watanabe, et al. Nutrient-dependent increased dendritic arborization of somatosensory neurons. *Genes Cells*, 2017, 22(1):105-14.).

(6) The sample size of some behavioural experiments are too low for this reviewer to be confident about the conclusions. These include Fig 3d/g/h and Fig 5c/d. In line 997, the authors state, "For all behavioural assays, each genotype was tested multiple times on different days, and data from all trials were combined." The authors should explain how they tested each genotype multiple times on different days and still only obtained 11-13 samples in Fig 5d.

(7) The sample sizes are particularly low for the experiment testing rolling probability in Fig 5d. This reviewer was curious how samples of such low sizes could give highly significant differences between the groups, so we did statistical tests based on authors' description of the sample sizes. Authors described in line 294-298 that "Similarly, refeeding following starvation significantly reduced rolling probability upon mechanonociceptive stimulation and optogenetic C4da activation (Fig. 5c; Fed, 64 %, n = 39; Starved, 62 %, n=37; Refed, 34%, n=35, and 5d; Fed, 42%, n=12; Starved, 46%, n=13; Refed, 18 %, n = 11)." Based on this description, the data for Fig 5d were calculated to be: Fed: 5 rolling, 7 no-rolling; Starved: 6 rolling, 7 no-rolling; and Refed: 2 rolling, 9 no-rolling. In contrast to authors' conclusions, neither Fisher's exact tests nor Chi-square tests yielded any significant differences between the groups. This raises the concern whether statistical analyses were done properly in this paper.

(8) In the calcium imaging results, the authors should explain how the ROIs were set. They are said to be set on the neurites (presumably the axon terminals) of C4da neurons, but this description isn't sufficient for either evaluating the validity of the results or the purpose of reproducibility. Were the ROIs encompass all the neurites of all C4da in a larva? Or, are they with one segment of the VNC? Are they selected randomly?

(9) More details should be provided in Methods describing Ca²⁺ imaging. Authors describe that larvae were pinned down on a dish and dissected along the dorsal midline. Internal organs except for neural tissues were removed and the VNC was imaged. Based on this description, larval muscle contractions would disrupt Ca²⁺ imaging of the central nervous system. This would cause variations in Ca²⁺ signal intensities. How did authors obtain stable Ca²⁺ imaging? If no measures were used to stabilize imaging, how did they decide which data to keep and which to discard? Do the sample numbers in the figures show the number of ROIs or larvae? The numbers seem to be low (in several experiments 4-7), especially if the samples are ROIs and there are multiple ROIs in one larva.

Minor concerns:

(10) In the abstract, 'yet how feeding evokes nociceptive suppression remains largely unknown' is

inaccurate. Much has been learned about this in other species.

(11) More details should be provided about manual quantification of behaviour. For example, How was the manual analysis done by using imageJ? What is the difference between using imageJ and just manual analysis?

(12) In Fig 4a and b, the control groups have very different Ca²⁺ levels (about 75% in Fig 4a versus ~250% in Fig 4b). Authors should explain this discrepancy.

(13) The sample numbers are missing in Fig 2b.

(14) The rolling probabilities for the control groups of mechanical stimulation are highly variable throughout the manuscript. In Fig 2c (36%), 2e (70%), and 3b (28%), the rolling probabilities differ dramatically. The authors should explain why the rolling probability is so variable in the control groups and whether this large variation affects their conclusions.

(15) In line 171, the authors state that "the control larvae continuously rolled and rarely stopped during the 35-s observation period." However, it is unclear which time period is the 35s. The authors explain in the figure legend that during the 15-s optogenetic stimulation, the larvae's behaviour was quantified. The authors should clarify this discrepancy.

(16) In the legend for Fig 2b, the * indicates $p < 0.005$, which is different from other figures. The authors should make this consistent with other figures or confirm if this is a typo.

(17) In line 390, "expression of InRDN lowered the Ca²⁺ levels of SDGs in sugar-refed larvae" there is a missing space between "SDGs" and "in".

(18) While the brain images in the top row of Fig 1c and Extended Fig 1a/b are about the same size, the scale bars showing 50 μ m in these two figures are of different sizes.

Reviewer #2 (Remarks to the Author):

In this manuscript, Nakamizo-Dojo et al. identified a neural pathway by which sugar sensing suppresses nociceptive response mediated by C4da neurons in *Drosophila* larvae. Specifically, using an approach that combines anatomical technique, behavior analysis, and Ca²⁺ imaging, the authors suggest that presence of ingested nutritive sugars is detected by CN neurons, which then trigger the insulin-producing neurons to release ILP2 – an insulin-like peptide – that then acts through InR to activate a group of descending GABAergic neurons (SDGs) that exert inhibition of nociceptive C4da neurons presynaptically. In general, I find the main message of this MS interesting, and the results generally well put together. However, there are a few issues I think the authors should address to strengthen their conclusions before publication.

First, for many key experiments described in this MS, a GAL4 only genotype control was missing. For example, in Figure 1d, the authors compared the rolling latency of SDGs-GAL4/Kir and +/Kir animals but did not include that from SDGs-GAL4/+ animals. The author should either include the SDGs-GAL4/+ only control or include a control that uses empty-split-GAL4s to drive the effectors of interest. This is not a trivial request as 1) recent work suggests that some of the commonly used insertion sites for these GAL4 lines (e.g., attp40) have an impact of neural function and/or development and 2) some of the manipulations reported here did not have a huge effect size to begin with.

Second, the results that suggest release of ILP2 – and its action on SDGs through InR – as the main driver by which sugar refeeding suppresses nociceptive response are not as convincing. To better solidify this point, the authors should 1) assess whether direct activation of CNs or ILP2 neurons in fed flies can recapitulate impact as sugar refeeding after starvation, and 2) use methods other than baseline GCaMP to assess the activity change in SDGs induced by manipulating InR, a molecule known to affect development of neurons. On a related note, the behavior effect of InR manipulation in SDGs should also be measured under conditions where InR transgenes expression is induced conditionally as opposed to chronically.

Third, it is not clear what activates SDGs in regular (non-starved) flies. SDGs must be active under regular state, else the authors would not have seen the effects of silencing SDGs as described on Figure 1-4. The signal(s) that activates SDGs in regular state must be conveyed to the SDGs differently from the one activated by refeeding as inactivating CNs or ILP2 neurons did not appear to affect fed flies' response to nociceptive stimuli (Figure 6-7). Can the authors inspect the connectome to provide some clues as to what are the neurons that directly synapse onto SDGs?

Lastly, it might be worth discussing if the suppression of nociceptive response the authors observed the truly the results of animals "prioritizing" feeding over pain or, alternatively, perhaps bingeing on sugar following starvation induces a state (e.g., sleep state) that suppresses movements in general. (One suppose if animals were to prioritize feeding when starved, the impact of nocifensive suppression should be present immediately upon contact with sugars as opposed to waiting for sugars to tripper ILP2 release.) On a related note, it might be interesting to test whether the nociception suppression upon refeeding of starved flies is specific to nutritive sugar, or could similar effects be observed when starved animals are refed with proteins and/or fats.

Reviewer #3 (Remarks to the Author):

The manuscript by Dojo et al is a stunning manuscript showing a descending nociceptive inhibitory circuit from the brain directly to pain sensing neurons in *Drosophila*. This cascade is initiated through sugar sensing during the refeeding period, including molecular interactions in the fly brain that initiate

the descending circuit. The authors use an array of behavioral, molecular, genetics, and imaging techniques with easy presentation and robust quantification to really tell a complete and impactful story. This work represents an important new addition to the emerging body of literature of descending pain inhibition across a wide variety of animals. This is one of the best manuscripts I have read in a long time and I cannot find any flaws in the work. One potential limitation is there is only a singular read-out for nociception, which is rolling behavior. I understand that this might be a limitation of the system and the relatively limited repertoire of behaviors that a fly larva can display. Perhaps adding place aversion experiments would be nice as another independent readout, but this is not necessary because this paper is already complete. I recommend publication without delay.

Thank you so much for the positive and constructive comments on our submission. According to the editor's and reviewers' comments, we have added multiple data and made changes to the text. We believe that our experimental additions substantially strengthen the major points of our manuscript, which was already favorably received by each reviewer. We hope that you will find our revised manuscript suitable for publication in *Nature Communications*.

Point-by-point responses to Reviewers' comments

Reviewer #1 (Remarks to the Author):

In this manuscript, Nakamizo-Dojo et al. report a neural mechanism by which feeding of starved Drosophila larvae overrides the escape behaviour elicited by noxious stimuli. They first report a cluster of GABAergic neurons in the larval brain that suppress nociceptive inputs through GABAB receptors in the nociceptors. They then found that glucose feeding to starved larvae activates these neurons through glucose-sensing neurons and insulin signaling. This attenuates the escape behaviour in response to noxious stimuli.

The discovery of the SDG descending GABAergic neurons in Drosophila is important. Although connectome studies have indicated the existence of descending neurons, physiological and functional investigations have not been done to establish the existence and functional significance of such neurons. The finding the sugar-sensing neurons in the brain inhibits SDG neurons is also novel and important, as it offers mechanistic insights into the suppression of nociception by feeding at the circuit level. While studies in mammals have shown that feeding suppresses nociceptive responses and uncovered the circuitry and molecular mechanisms, this study is the first in Drosophila, which is still novel and significant.

Overall, the studies are well designed (though see concerns below) and the paper well written. The use of multiple ways to stimulate the nociceptors strengthens the study and demonstrates the regulation is across nociceptive modalities. However, there are many technical concerns.

Major concerns:

(1) To inhibit neurons, authors expressed Kir2.1 in target neurons throughout development. Because inhibiting neurons during development may change neuronal activity and circuitry, acute inhibition should be used in at least some key experiments. Such approaches are available and used in Drosophila, for example, GtACR. Reviewer recognizes that some experiments in this paper may not be possible to use GtACR, but some are. For example, the behaviour experiments.

We agree that this is an important point. To minimize developmental effects of neuronal silencing, we used GtACR1, a light-gated anion channel suggested by the reviewer, for temporal silencing of SDGs and CN neurons. As a result, larvae expressing GtACR1 in SDGs showed a significantly higher level of rolling probability when mechanical stimuli were applied during 10 s of green-light illumination, compared to the genetic controls and the no-light group (**Fig. 2e**). Moreover, optogenetic inhibition of *GtACR1*-expressing CN neurons during the 1-h refeeding period cancelled the sugar-induced rolling delay (**Fig. 6d**). Since these results from GtACR1-mediated acute silencing were consistent with those from chronic silencing by Kir2.1 (**Figs. 2d** and **6c**), we concluded that the behavioral phenotypes observed are least likely due to developmental perturbations. We added these GtACR1 experiments to the main figures (**Figs. 2e** and **6d**) and described them in the text (p. 7–8, lines 147–153; p.17 lines 372–376).

(2) Evidence showing that Gad1 knockdown reduces Gad1 expression is needed. If a Gad1 antibody isn't available, GABA staining should be used to test Gad1 knockdown. Moreover, one RNAi line is insufficient to establish the function of a gene. Additional RNAi lines, rescue with transgenes, or genetic mutants should be used to corroborate the claimed functions of Gad1 and GABABR1/R2.

Thank you for pointing out those important genetic issues. According to the suggestions, we conducted the following experiments to confirm the effect of RNAi.

(i) Since the anti-Gad1 antibody was unavailable, we performed immunostaining against GABA as suggested by the reviewer. Signal intensity at the cell bodies of SDGs was significantly reduced in the *Gad1* RNAi group, as attached below.

GABA synthesis was significantly reduced by *Gad1* knockdown in SDGs.

Immunoreactivity of GABA at the cell bodies of SDGs was normalized to the median value of the no-UAS control. The thick line and the thin error bar represent the median and the interquartile range, respectively. ** $p < 0.005$ (Mann–Whitney U-test).

(ii) We carried out the same set of behavioral experiments using another RNAi lines for *Gad1* and *GABA-B-R1/2*, which all led to consistent results with the first RNAi lines. Similar to the first RNAi line for *Gad1* (*TRiP.HMC03350* in attP40 (BL#51794)), additional two lines (*TRiP.JF02916* in attP2 (BL#28079) and *GD8508* (VDRC#32344)) induced a battery of phenotypes indicative of SDGs malfunctioning; increased rolling probability by mechanical stimulation (**Extended Data Fig. 3e**), shortened rolling latency after noxious heat application (**Extended Data Fig. 3f**), and prolonged duration of rolling triggered by C4da optogenetic stimulation (**Extended Data Fig. 3g**). In parallel, the initial RNAi experiments for *GABA-B-R1* (*TRiP.HMC03388* in attP2 (BL#51817)) and *R2* (*TRiP.HMC02975* in attP2 (BL#50608)) were phenocopied by independent RNAi elements (*KK109166* in VIE260b (VDRC#101440) for R1; *KK100020* in VIE260b (VDRC#110268) for R2) expressed in C4da; increased tendency in the mechanically-induced rolling probability (**Extended Data Fig. 3l**) and extended rolling upon C4da optogenetic activation (**Extended Data Fig. 3m**). We added these new data to the revised manuscript (p. 10, lines 210–213; p. 11, lines 234–238) as **Extended Data Figs. 3e–g** and **3l–m**.

(3) *Negative controls for GABA and Gad1 staining in Fig 3a are missing.*

We thank the reviewer for raising these points. To confirm the specificity of signal detection, the following negative controls were included in the revised version as

Extended Data Fig. 3b.

(i) When GABA staining was performed without adding the anti-GABA primary antibody to the brain specimen, the signal at the SDGs cell body was under the detection level (**Extended Data Fig. 3b, top**). The specificity of GABA immunoreactivity was also supported by the reduced signal detected with the same antibody when *Gad1* was knocked down (see comment (2) from the reviewer #1).

(ii) In the original figure, *Gad1*-positive cells were labeled by expressing *LexAop-GCaMP7s* via *Gad1-LexA*, a Trojan gene-trap line with *LexA* inserted into the *Gad1* locus (BL#60324), and the native fluorescence of GCaMP was directly observed without immunostaining. We thus crossed parental flies without *Gad1-LexA* as a “no-*LexA*” control to observe co-labeling with *SDGs-spGAL4*-driven *CsChrimson::mCherry*. As expected, the GCaMP native fluorescence in the no-*LexA* control (+>*GCaMP*) was not detected (**Extended Data Fig. 3b, bottom**), denying the possibility of leaky expression or background noise detection.

(4) The evidence that CN neurons mediate refeeding-induced nociceptive suppression is weak. Data showing that CN neuron activity changes in the Refed group is missing. Moreover, if authors' model is correct, then activating CN neurons should inhibit nociceptive behaviour and C4da responses.

We are grateful for the reviewer's valuable suggestions. Two issues, (i) measurement of neuronal activity in CN neurons and (ii) behavioral and physiological consequences of forced activation in CN neurons, have been separately addressed.

(i) Due to the technical hindrance for *ex vivo* Ca²⁺ imaging of CN neurons, we turned to the CaLexA (calcium-dependent nuclear import of LexA) system to assess the neuronal activity. Likewise the previous report in adult flies (Oh Y, *et al.* 2019 *Nature*), we expressed CaLexA in larval CN neurons and confirmed that signal intensity increased by sugar refeeding. We added this new data as **Extended Data Fig. 4e** described in the results (p. 17, lines 364–366) and the methods (p. 51–52, lines 1196–1203) sections.

(ii) We agree that activation experiments for CN neurons would definitively strengthen our model. In the revised manuscript, we performed both chronic and temporal activation of CN neurons through NaChBac expression and optogenetic stimulation, respectively. Based on our working hypothesis, we expected that direct activation of CN neurons could

mimic the analgesic effect of sugar refeeding. Indeed, NaChBac expression in CN neurons extended the rolling latency upon thermo-nociception in normally fed larvae (**Fig. 6e**). Moreover, optogenetic stimulation of CN neurons for 1 h following the 12-h starvation (**Fig. 6f**; note that larvae were not refed) effectively suppressed the thermo-nociceptive behavior (**Fig. 6g**). This temporal CN activation without refeeding was effective against C4da responses as well; the 1-h optogenetic stimulation of CN neurons prior to imaging partly suppressed the optogenetically induced calcium influx in C4da (**Fig. 6h**; note that CN and C4da expressed different channelrhodopsins, red-light responsive CsChrimson and blue-light responsive ChR2, enabling independent stimulation). These results further support that CN neurons mediate the sugar refeeding-induced nociceptive suppression (**Fig. 6i**). We added these new data to the main figures (**Fig. 6e–h**) and described them in the text (p. 17, lines 376–388).

(5) The proper control for the refed experiment should be sucrose-soaked paper only. The addition of yeast paste complicates the experiment, especially considering the report that dendrites of C4da neurons hyperarborize on a low-yeast diet (Watanabe, et al. Nutrient-dependent increased dendritic arborization of somatosensory neurons. Genes Cells, 2017, 22(1):105-14.).

Thank you for the suggestion. As larvae are deprived of yeast during starvation and sugar-refeeding, the fed state supplied with “yeast plus sucrose” clearly differs in accessibility to yeast diet compared with the other two states. The previous report mentioned by the reviewer demonstrated that long-term exposure, throughout the larval developmental period, to a low-yeast diet affects the C4da arborization (Watanabe, et al. 2017 *Gene Cells*). We at least confirmed that larvae fed “yeast plus sucrose” and “sucrose only (*i.e.*, yeast-deprived)” for 6 h were statistically indistinguishable in terms of rolling latency upon thermo-nociception (as shown below), suggesting that yeast deprivation under our experimental conditions less likely impact the rolling latency. Although this does not exclude, if any, the potential effect of yeast diet on the nociceptive control, we decided to leave it to future studies and herein keep the “yeast plus sucrose” feeding procedure to achieve the “fed” state of larvae.

Larvae fed sucrose with or without yeast show comparable thermonociceptive response.

Larvae raised to early 3rd instar in standard food were collected and placed on a sucrose-soaked paper with or without yeast paste. After 6 h of feeding, larvae were subjected to heat assay. Rolling latency after the heat probe attachment was plotted. The thick line and the thin error bar represent the median and the interquartile range, respectively. n.s. $p \geq 0.05$ (Mann–Whitney U-test).

(6) *The sample size of some behavioural experiments are too low for this reviewer to be confident about the conclusions. These include Fig 3d/g/h and Fig 5c/d. In line 997, the authors state, "For all behavioural assays, each genotype was tested multiple times on different days, and data from all trials were combined." The authors should explain how they tested each genotype multiple times on different days and still only obtained 11-13 samples in Fig 5d.*

(7) *The sample sizes are particularly low for the experiment testing rolling probability in Fig 5d. This reviewer was curious how samples of such low sizes could give highly significant differences between the groups, so we did statistical tests based on authors' description of the sample sizes. Authors described in line 294-298 that "Similarly, refeeding following starvation significantly reduced rolling probability upon mechanonociceptive stimulation and optogenetic C4da activation (Fig. 5c; Fed, 64 %, $n = 39$; Starved, 62 %, $n=37$; Refed, 34%, $n=35$, and 5d; Fed, 42%, $n=12$; Starved, 46%, $n=13$; Refed, 18 %, $n = 11$)." Based on this description, the data for Fig 5d were calculated to be: Fed: 5 rolling, 7 no-rolling; Starved: 6 rolling, 7 no-rolling; and Refed: 2 rolling, 9 no-rolling. In contrast to authors' conclusions, neither Fisher's exact tests nor Chi-square tests yielded any significant differences between the groups. This raises the concern whether statistical analyses were done properly in this paper.*

We agree that more attention should have been paid to the sample sizes. To objectively set the sample sizes with statistical reliability, we applied power analyses to Figs. 3d, 3g,

3h, 5c, and 5d (figure numbers in the initial manuscript).

The “power” (= $1 - \beta$ error probability) was calculated by post-hoc analyses based on the actual sample numbers used in the initial manuscript. As a result, the power values for Figs. 3d, 3g, and 3h were calculated to be > 0.90 (see the blue-colored row in the attached Table 1). Since power is conventionally set at 0.80 (Banerjee A, et al. 2009), we decided that these 3 figures had enough power to draw our conclusion.

Table 1. Post-hoc power analyses of Figs. 3d, 3g, and 3h from the initial manuscript

Test family Statistical test Type of power analysis Tail(s) Parent distribution	t tests Mears-Wilcoxon-Mann-Whitney test (two groups) Posthoc: Compute achieved power - given α , sample size, and effect size			
	Two			
	Normal			
	Figure number (comparison groups)	Fig. 3d	Fig. 3g (left vs middle)	Fig. 3g (left vs right)
Mean group1	3.5	6.4	6.4	7.225
Mean group2	10.34615385	48.55	20.4	55.97368421
SD α group1	2.154729018	5.524109835	5.524109835	8.906642997
SD α group2	4.03311612	16.70085091	16.9392102	12.73934597
Effect size d	2.117367	3.388663	1.111229	4.435189
α err prob	0.05	0.05	0.05	0.05
Sample size group1	15	20	20	20
Sample size group2	13	20	20	19
Power ($1 - \beta$ err prob)	0.9994784	1	0.9164382	1

Statistical tests applied in Figs. 5c and 5d (Fisher’s exact test) were different from those in Figs. 3d, 3g, and 3h (Mann–Whitney test). We thus performed post-hoc power analyses in a similar manner but separately. As shown in the attached Table 2, the power values for Figs. 5c and 5d from the initial manuscript ranged from 0.15 to 0.69. As the reviewer anticipated, we considered these 2 experiments need larger sample sizes to avoid the type II error (false negative).

Table 2. Post-hoc power analyses of Figs. 5c and 5d based on old dataset in the initial manuscript

Test family Statistical test Type of power analysis Tail(s) Parent distribution	Exact Proportions: inequality, two independent groups (Fisher’s exact test) Posthoc: Compute achieved power - given α , sample size, and effect size			
	Two			
	Normal			
	Figure number (comparison groups)	Fig. 5c (left vs right)	Fig. 5c (middle vs right)	Fig. 5d (left vs right)
Proportion p1	0.64	0.62	0.42	0.46
Proportion p2	0.34	0.34	0.18	0.18
Ratio P1 /P2	1.882353	1.823529	2.333333	2.555556
α err prob	0.05	0.05	0.05	0.05
Sample size group1	39	37	12	13
Sample size group2	35	35	11	11
Power ($1 - \beta$ err prob)	0.6884536	0.608231	0.1506002	0.2586712

To determine appropriate sample sizes that provide higher power than the standard 0.80, we performed *a priori* power analyses based on the initial dataset of Figs. 5c and 5d. The desired sample number for each group was estimated to be 49–63 (see the orange-colored

rows in the attached Table 3), aiming the power of 0.80.

Table 3. *A priori* power analyses of Figs. 5c and 5d based on old dataset in the initial manuscript

Test family Statistical test Type of power analysis Tail(s)	Exact Proportions: Inequality, two independent groups (Fisher's exact test) A priori: Compute required sample size - give α , power, and effect size			
	Two			
Figure number (comparison groups)	Fig. 5c (left vs right)	Fig. 5c (middle vs right)	Fig. 5d (left vs right)	Fig. 5d (middle vs right)
Proportion p1	0.64	0.62	0.42	0.46
Proportion p2	0.34	0.34	0.18	0.18
Ratio P1/P2	1.882353	1.823529	2.333333	2.555556
α err prob	0.05	0.05	0.05	0.05
Power (1- β err prob)	0.8	0.8	0.8	0.8
Allocation ratio N2/N1	1	1	1	1
Sample size group1	49	57	63	49
Sample size group2	49	57	63	49
Total sample size	98	114	126	98

According to the above estimation, we newly repeated the same experiments with more larvae (≥ 70 for each group). The overall patterns were highly reproducible. As a final check, we performed post-hoc analyses once again on the new dataset. The power values became > 0.90 , surpassing the conventional level (see the blue-colored row in the attached Table 4). We thus replaced these two figures (**Figs. 5c and 5d**) in the revised manuscript.

Table 4. Post-hoc power analyses of Figs. 5c and 5d using new dataset in the revised manuscript

Test family Statistical test Type of power analysis Tail(s) Parent distribution	Exact Proportions: Inequality, two independent groups (Fisher's exact test) Post-hoc: Compute achieved power - given α , sample size, and effect size			
	Two Normal			
Figure number (comparison groups)	Fig. 5c (left vs right)	Fig. 5c (middle vs right)	Fig. 5d (left vs right)	Fig. 5d (middle vs right)
Proportion p1	0.65	0.57	0.57	0.54
Proportion p2	0.34	0.34	0.28	0.28
Ratio P1/P2	1.911765	1.676471	2.035714	1.928571
α err prob	0.05	0.05	0.05	0.05
Sample size group1	115	70	74	80
Sample size group2	117	70	79	79
Power (1- β err prob)	0.9974566	0.9445529	0.948096	0.9029414

In addition, the following statement in the methods section was omitted: *"For all behavioral assays, each genotype was tested multiple times on different days, and data from all trials were combined."*

(8) *In the calcium imaging results, the authors should explain how the ROIs were set. They are said to be set on the neurites (presumably the axon terminals) of C4da neurons, but this description isn't sufficient for either evaluating the validity of the results or the*

purpose of reproducibility. Were the ROIs encompass all the neurites of all C4da in a larva? Or, are they with one segment of the VNC? Are they selected randomly?

We thank the reviewer for raising this point. We observed the A5–6 segments in the VNC, and 4 ROIs, each with a diameter of 1 μm per one segment of either side (depicted in the schema of Fig. 1a as the triangle-shaped region of C4da axon terminal), were set. After the mean signal intensity within each ROI was measured using ImageJ, values from 4 ROIs were averaged to represent the data of one brain sample. We added the description of ROI setting in the methods section (p. 50, lines 1160–1163).

(9) More details should be provided in Methods describing Ca²⁺ imaging. Authors describe that larvae were pinned down on a dish and dissected along the dorsal midline. Internal organs except for neural tissues were removed and the VNC was imaged. Based on this description, larval muscle contractions would disrupt Ca²⁺ imaging of the central nervous system. This would cause variations in Ca²⁺ signal intensities. How did authors obtain stable Ca²⁺ imaging? If no measures were used to stabilize imaging, how did they decide which data to keep and which to discard? Do the sample numbers in the figures show the number of ROIs or larvae? The numbers seem to be low (in several experiments 4-7), especially if the samples are ROIs and there are multiple ROIs in one larva.

Related to comment #8, we agree that the details of Ca²⁺ imaging procedures need to be clarified. As the reviewer anticipated, we had initially noticed that the brain frequently moved due to larval muscle contractions when the *ex vivo* preparation was soaked in a buffer containing Ca²⁺ ions. We thus modified the buffer recipe to be “calcium-free” (70 mM NaCl, 5 mM KCl, 4 mM MgCl₂, 10 mM NaHCO₃, 5 mM trehalose, 115 mM sucrose, 5 mM HEPES, pH 7.2) to avoid the unwanted movements from muscle contractions. Although not as frequently and intensively as above, we still experienced subtle drifting of the brain in some cases. To further obtain stable Ca²⁺ signals, each image was manually checked and motion correction was applied using the StackReg/TurboReg plugins (P. Thévenaz, *et al.* 1998 *IEEE Trans Image Process*; available at <http://bigwww.epfl.ch/thevenaz/stackreg/>) for ImageJ. These technical efforts enabled us to use virtually all images captured for analyses without discarding any data. The (n) values shown within figures indicate the number of larval brain preps (not the number of ROIs) for each genotype/treatment group. Together with the details of ROI setting, we added the above information to the methods section (p. 49–50, lines 1140–1168).

Minor concerns:

(10) *In the abstract, ‘yet how feeding evokes nociceptive suppression remains largely unknown’ is inaccurate. Much has been learned about this in other species.*

We agree that this sentence needs to be more accurate. As the reviewer mentioned, many observations have been made in a wide variety of animals where feeding suppresses nociceptive escape. At the neural circuit level, mammalian models have demonstrated the involvement of several local interneurons and descending pathways. In terms of molecular mechanisms, not only GABA but multiple neurotransmitters (*e.g.*, serotonin, noradrenaline, enkephalin, etc.) act in concert to mediate the nociceptive modulation, and the sugar-inducible insulin is found in the mammalian brain. So far, what has not been fully done is to connect these fragmented pieces of information within a single model. We believe that the fruit fly larva is one such model that enables us to quantitatively analyze the mechanisms of feeding-induced nociceptive suppression at the levels of molecule, neural circuitry, and behavior. We thus replaced the sentence as follows: “..., yet underlying molecular and cellular mechanisms are incompletely understood.” (p. 2, lines 22–23)

(11) *More details should be provided about manual quantification of behaviour. For example, How was the manual analysis done by using imageJ? What is the difference between using imageJ and just manual analysis?*

Thank you for the suggestion. The initial description (“all movies were manually analyzed with ImageJ to measure the larval rolling duration and/or latency”) needs to be corrected. For behavioral assays, all movies were manually checked frame-by-frame. In this regard, ImageJ was only used to load the movies (saved as avi or multi-tiff files), consisting of series of images captured at certain frequencies, to observe each frame without any special processing. Manual detection of the rolling initiation, defined as a “corkscrew-like” behavior in which a larva turned on its long axis more than 90°, was followed by quantification of rolling parameters. Duration and latency were measured based on time stamps extracted from recorded movies using the VLC media player (available at https://www.videolan.org/vlc/index.en_GB.html) with an extension plugin “Time v3.2”

(available at <https://addons.videolan.org/p/1154032/>). We added these technical details to the methods section (p. 47, lines 1085–1094).

(12) In Fig 4a and b, the control groups have very different Ca²⁺ levels (about 75% in Fig 4a versus ~250% in Fig 4b). Authors should explain this discrepancy.

Calcium imaging experiments were performed under distinct conditions optimized for each purpose. For example, **Fig. 4a** was recorded at a gain level of “Gain 1x, EM Gain 150” and an exposure window of 100 ms, whereas **Fig. 4b** was imaged at “Gain 2x, EM Gain 100” with a 200-ms exposure. To avoid confusion, detailed conditions for each experiment are listed in the **Supplementary Table 2**, and the following sentence is added to the legend of Fig. 4: “Note that each experiment was carried out under distinct imaging condition optimized for each purpose. See Supplementary Table 2 for detailed imaging conditions.”

(13) The sample numbers are missing in Fig 2b.

Thank you for catching this mistake. We added the number of larvae tested at each LED condition as “(n) = 7–10” in the revised version of **Fig. 2b**.

(14) The rolling probabilities for the control groups of mechanical stimulation are highly variable throughout the manuscript. In Fig 2c (36%), 2e (70%), and 3b (28%), the rolling probabilities differ dramatically. The authors should explain why the rolling probability is so variable in the control groups and whether this large variation affects their conclusions.

We are grateful that the reviewer’s sharp observation made us realize some technical and genetic issues that had been overlooked.

After a close validation of the experimental procedures, we found out that agarose substrates used in mechanonociception assays were not unified throughout the manuscript; it was 8 ml of 1% agarose in **Fig. 2d** (Fig. 2c in the initial manuscript) while 12 ml of 2% agarose (as described in the methods section) was used in all other experiments. We had been using 1% agarose at the early stage, but during the course of

this study we changed the recipe to 2% according to another report (Hoyer, *et al.* 2018 *Bio Protoc*); we had empirically found that rolling probability of larvae placed on 1% agarose was often lower than that on 2% agarose. To make it consistent across the presented figures, we re-performed the entire experiment for Fig. 2d with 2% agarose substrates. As a result, an overall increase was observed in the rolling probability (**Fig. 2d**; No-*GAL4* control, 56%, n = 98; No-*UAS* control, 52%, n = 100; SDGs silencing, 81%, n = 100).

We next sought to determine the reason why the initial version of Fig. 3b showed such a low level of rolling probability. One critical difference between the No-*UAS* control in Fig. 3b and that in other figures is that $y[1] \ v[1]; P\{y[+t7.7]=CaryP\}attP40$ (BL#36304), a control strain harboring the attP landing site originally used to create the TRiP RNAi collection, was crossed with *SDGs-spGAL4* to obtain offspring larvae. We suspected that the genetic background of BL#36304 might have affected the results. If so, the test group (*SDGs-spGAL4*>*UAS-IR-Gad1*) might be similarly affected since the parental *UAS-IR-Gad1* line, $y[1] \ v[1]; P\{y[+t7.7] \ v[+t1.8]=TRiP.HMC03350\}attP40$ (BL#51794), mostly share the genetic background with BL#36304. It turned out to be the case, as simply crossing $w[1118]$ with either BL#36304 or BL#51794 had a significant impact on the rolling probability in tested offspring, as attached below.

Genetic backgrounds of BL#36304 and BL#51794 affect larval rolling induced by mechanical stimulation.

Offspring larvae obtained from indicated crosses were subjected to mechanical stimulation.

BL#36304 ($y[1] \ v[1]; P\{y[+t7.7]=CaryP\}attP40$): the control strain harboring the attP landing site originally used to create the TRiP RNAi collection.

BL#51794 ($y[1] \ v[1]; P\{y[+t7.7] \ v[+t1.8]=TRiP.HMC03350\}attP40$): the *UAS-IR-Gad1* line used for *Gad1* RNAi in Fig. 3b.

* p < 0.05 (Mann–Whitney U-test with Bonferroni correction)

Ideally the genetic elements could be purified by outcrossing several (conventionally 5 or 6) generations, but for the sake of time, we replaced X and 3rd chromosomes of BL#36304 and BL#51794 with those derived from *w[1118]* used in other figures as parents. In this way, the 2nd chromosome harboring the *CaryP* attP40 landing site or the *UAS-IR-Gad1* element is still unchanged, while the direct/indirect effects from X and 3rd chromosomes could theoretically be reduced. As a result, both groups of larvae obtained from parents after replacing the chromosomes showed a slightly higher level of rolling probability. Importantly, the effect of *Gad1* RNAi in SDGs was still observed (**Fig. 3b**; No-*GAL4* control, 40%, n = 70; *Gad1*-RNAi in SDGs, 64%, n = 58). We thus replaced the dataset for **Fig. 3b** in the revised manuscript.

The exact cause of the reduced rolling probability by using these RNAi lines is currently unclear. Seemingly, the genetic background affected the probability of mechanonociceptive rolling more severely than escape phenotypes induced by other modalities such as heat (Fig. 3c) and C4da optogenetic stimulation (Fig. 3d). Moreover, the effect seems specific for the attP40-based RNAi lines; the strains used for *GABA_B-Rs* RNAi in Fig. 3f have their IR elements inserted in the attP2 site of 3rd chromosome, and the corresponding control line is *y[1] v[1]; P{y[+t7.7]=CaryP}attP2* (BL#36303), not BL#36304. A recent study actually raised potential issues for attP40-based transgenic lines, as the attP40 docking site is located within the *Msp300* gene encoding a Nesprin-like adapter protein, and thus its function is likely disrupted by the attP40-targeted insertion (van der Graaf K, *et al.* 2022 *PLoS ONE*). Another group reported an example where attP40 insertions somehow interfere with the functions of specific GAL4 drivers (Duan Q, *et al.* 2023 *G3 Bethesda*). Although we decided not to include lengthy explanations for these puzzling situations to the main text, we mentioned the parental genotypes in the figure legend of Fig. 3b as follows: “Note that the background genotypes of parental flies used in Fig. 3b were different from other experiments; see Supplementary Table 1 for full genotypes”.

(15) In line 171, the authors state that "the control larvae continuously rolled and rarely stopped during the 35-s observation period." However, it is unclear which time period is the 35s. The authors explain in the figure legend that during the 15-s optogenetic stimulation, the larvae's behaviour was quantified. The authors should clarify this discrepancy.

Thank you for catching this discrepancy. As described in the legend, the stop probability

in **Fig. 2h** (Fig. 2g in the initial version) was literally measured from the 15-s window of optogenetic stimulation. On the other hand, the “35-s observation period” was the time window from which raster plots were drawn in **Extended Data Fig. 2e**, consisting of 10 s pre-stimulation (no LED) + 15 s stimulation with LED + 10 s post-stimulation (no LED). To avoid confusions, we modified the schema and the legend of **Extended Data Fig. 2e** to explain the 35-s window represented in raster plots. We also changed the main text as “the control larvae continuously rolled and rarely stopped.” (p. 9, line 177).

*(16) In the legend for Fig 2b, the * indicates $p < 0.005$, which is different from other figures. The authors should make this consistent with other figures or confirm if this is a typo.*

We appreciate the comment about asterisks (*). Actually, it was not a typo and we had to make the asterisks small simply because of space constraints in **Fig. 2b**, but we agree that inconsistent usage of * within the figure could confuse the readers. In the revised version, we placed the asterisks in a vertical direction so that *** and ** fit into the space. The meaning of each asterisk is now consistent across the panels, as we stated in the legend: “*** $p < 0.0005$, ** $p < 0.005$ ”.

(17) In line 390, "expression of InRDN lowered the Ca²⁺ levels of SDGs in sugar-refed larvae" there is a missing space between "SDGs" and "in".

Thank you for catching this typo. We fixed it to “SDGs in” (p. 20, line 454).

(18) While the brain images in the top row of Fig 1c and Extended Fig 1a/b are about the same size, the scale bars showing 50 μ m in these two figures are of different sizes.

We appreciate the reviewer for catching this error. It seems that some of the scale bars had been unintentionally changed during the course of image enlargement/shrinkage. We closely verified the raw data for all confocal images including **Fig. 1c**, **Extended Data Fig. 1a** and **1b**, and corrected the scale bars. Changes made in the revised manuscript are summarized in the following table.

Figure number	Fig. 1a	Fig. 1c, top	Fig. 1c, middle	Fig. 1c, bottom	Fig. 1d	Fig. 1e	Fig. 2a	Fig. 2b	Fig. 4g	Fig. 7a
Label upon scale bar	10 µm	50 µm	5 µm	10 µm	50 µm	50 µm	10 µm	10 µm	10 µm	25 µm
Scale bar length on printed paper (mm)	Initial 4.88	Revised 1.62	Unchanged	Initial 6.837	Revised 2.28012	Initial 2230	Revised 6.776	Unchanged	Unchanged	Unchanged
	Initial 0.186	Revised 1.46	Initial 3.433	Revised 2.212	Unchanged	Unchanged	Unchanged	Unchanged	Unchanged	Unchanged

Figure number	Revised Data Fig. 1a, top		Revised Data Fig. 1a, middle		Revised Data Fig. 1a, bottom		Revised Data Fig. 1b, top		Revised Data Fig. 1b, middle		Revised Data Fig. 1b, bottom		Revised Data Fig. 2a	Revised Data Fig. 2b	Revised Data Fig. 2c
Label upon scale bar	50 µm		5 µm		50 µm		50 µm		5 µm		10 µm		50 µm	10 µm	50 µm
Scale bar length on printed paper (mm)	Initial 4.259	Revised 2.1405	Initial 7.286	Revised 1.625	Initial 2.764	Revised 7.286	Initial 4.259	Revised 2.2	Initial 7.286	Revised 2.916	Initial 2.764	Revised 7.671	Unchanged	Unchanged	Unchanged

Reviewer #2 (Remarks to the Author):

In this manuscript, Nakamizo-Dojo et al. identified a neural pathway by which sugar sensing suppresses nociceptive response mediated by C4da neurons in Drosophila larvae. Specifically, using an approach that combines anatomical technique, behavior analysis, and Ca²⁺ imaging, the authors suggest that presence of ingested nutritive sugars is detected by CN neurons, which then trigger the insulin-producing neurons to release ILP2 – an insulin-like peptide – that then acts through InR to activate a group of descending GABAergic neurons (SDGs) that exert inhibition of nociceptive C4da neurons presynaptically. In general, I find the main message of this MS interesting, and the results generally well put together. However, there a few issues I think the authors should address to strengthen their conclusions before publication.

1

First, for many key experiments described in this MS, a GAL4 only genotype control was missing. For example, in Figure 1d, the authors compared the rolling latency of SDGs-GAL4/Kir and +/Kir animals but did not include that from SDGs-GAL4/+ animals. The author should either include the SDGs-GAL4/+ only control or include a control that uses empty-split-GAL4s to drive the effectors of interest. This is not a trivial request as 1) recent work suggests that some of the commonly used insertion sites for these GAL4 lines (e.g., attp40) have an impact of neural function and/or development and 2) some of the manipulations reported here did not have a huge effect size to begin with.

We thank the reviewer for raising this important genetic issue. Although “Figure 1d” had pointed out in this reviewer’s comment, we assume it was supposed to be Fig. 2d because Figure 1 contained brain images with no behavioral results. As Figure 2 includes key experiments demonstrating the inhibitory role of SDGs on nociceptive behavior for the first time, we added the GAL4 only (no-UAS) control to the following panels in the revised manuscript: **Figs. 2a** (SDGs silencing with C4da optogenetic stimulation), **2c** (SDGs silencing with thermal stimulation; Fig. 2d in the initial version), **2d** (SDGs silencing with mechanical stimulation; Fig. 2c in the initial version), **2f** (SDGs activation with mechanical stimulation; Fig. 2e in the initial version). Differences between the test group (SDGs-GAL4>UAS-Kir2.1) and the newly added no-UAS control were statistically significant in all of these figures. Note that the original dataset of **Fig. 2d** (SDGs silencing with mechanical stimulation; Fig. 2c in the initial version) was entirely replaced after re-performing the experiment, according to another reviewer’s comments (see reviewer #1-

comment #14).

Regarding the sample size, we performed post-hoc power analyses for the above figures to confirm the statistical reliability of the presented dataset. The power values ($= 1 - \beta$ error probability) were > 0.97 (see the blue-colored row in the attached Table 5 and 6), surpassing the conventional value of 0.80 (Banerjee A, et al. 2009).

Table 5. Post-hoc power analyses of Figs. 2a and 2c using new dataset in the revised manuscript

Test family Statistical test Type of power analysis Tail(s) Parent distribution	t tests			
	Mears-Wilcoxon-Mann-Whitney test (two groups)			
	Post hoc: Compute achieved power - given α , sample size, and effect size			
	Two Normal			
Figure number (comparison groups)	Fig. 2a (left vs right)	Fig. 2a (middle vs right)	Fig. 2c (left vs right)	Fig. 2c (middle vs right)
Mean group1	1.678571429	2.134615385	3.993014286	4.632571429
Mean group2	10.56	10.56	2.2705	2.2705
SD σ group1	0.696340513	1.323311697	1.917597844	2.161342367
SD σ group2	5.202803731	5.202803731	1.065403071	1.065403071
Effect size d	2.392793	2.219501	1.11046	1.386281
α err prob	0.05	0.05	0.05	0.05
Sample size group1	14	28	70	70
Sample size group2	25	25	70	70
Power(1- β err prob)	0.9999994	1	0.9999949	1

Table 6. Post-hoc power analyses of Figs. 2d and 2f using new dataset in the revised manuscript

Test family Statistical test Type of power analysis Tail(s) Parent distribution	Exact			
	Proportions: Inequality, two independent groups (Fisher's exact test)			
	Post hoc: Compute achieved power - given α , sample size, and effect size			
	Two Normal			
Figure number (comparison groups)	Fig. 2d (left vs right)	Fig. 2d (middle vs right)	Fig. 2f (left vs right)	Fig. 2f (middle vs right)
Proportion p1	0.56	0.52	0.7	0.6
Proportion p2	0.81	0.81	0.16	0.16
Ratio P1/P2	0.691358	0.6419753	4.375	3.75
α err prob	0.05	0.05	0.05	0.05
Sample size group1	98	100	50	50
Sample size group2	100	100	50	50
Power (1- β err prob)	0.9655725	0.9904622	0.9999152	0.9956051

2

Second, the results that suggest release of ILP2 – and its action on SDGs through InR – as the main driver by which sugar refeeding suppresses nociceptive response are not as convincing. To better solidify this point, the authors should 1) assess whether direct activation of CNs or ILP2 neurons in fed flies can recapitulate impact as sugar refeeding after starvation, and 2) use methods other than baseline GCaMP to assess the activity change in SDGs induced by manipulating InR, a molecule known to affect development of neurons. On a related note, the behavior effect of InR manipulation in SDGs should also be measured under conditions where InR transgenes expression is induced conditionally as opposed to chronically.

Thank you very much for the constructive suggestions. The following three issues have been addressed: (i) behavioral and physiological consequences of forced activation in CN neurons, (ii) measurement of SDGs neuronal activity using methods other than GCaMP, and (iii) genetic manipulation of InR in a temporally controlled manner.

(i) We agree that activation experiments for CN neurons would definitively strengthen our model. In the revised manuscript, we performed both chronic and temporal activation of CN neurons through NaChBac expression and optogenetic stimulation, respectively. Based on our working hypothesis, we expected that direct activation of CN neurons could mimic the analgesic effect of sugar refeeding. Indeed, NaChBac expression in CN neurons extended the rolling latency upon thermo-nociception in normally fed larvae (**Fig. 6e**). Moreover, optogenetic stimulation of CN neurons for 1 h following the 12-h starvation (**Fig. 6f**; note that larvae were not refed) effectively suppressed the thermo-nociceptive behavior (**Fig. 6g**). This temporal CN activation without refeeding was effective against C4da responses as well; the 1-h optogenetic stimulation of CN neurons prior to imaging partly suppressed the optogenetically induced calcium influx in C4da (**Fig. 6h**; note that CN and C4da expressed different channelrhodopsins, red-light responsive CsChrimson and blue-light responsive ChR2, enabling independent stimulation). These results further support that CN neurons mediate the sugar refeeding-induced nociceptive suppression (**Fig. 6i**). We added these new data to the main figures (**Fig. 6e–h**) and described them in the text (p. 17, lines 376–388).

(ii) To examine the SDGs neuronal activity, we applied CaLexA and CAMPARI2 methods to starved/refed larvae. As attached below, however, the signals from CaLexA and CAMPARI2 reporters were rather weak and inconsistent. This was probably due to the weak driving force of the split GAL4 and/or the reporter element itself (CaLexA consists of *LexAop-CD8-GFP-2A-CD8-GFP*; *UAS-mLexA-VP16-NFAT*, *LexAop-rCD2-GFP* and CAMPARI2 is provided as *UAS-CaMPARI2* in VK00005) with apparently less copies of the UAS/LexAop sites compared to GCaMP (*20XUAS-IVS-jGCaMP7s* in su(Hw)attP5). Nonetheless of the unstable baseline signals, we pushed ourselves to test whether sugar refeeding could induce any detectable changes. Although a small portion of cells (3 out of 16) showed bright CaLexA signals with a > 2-fold intensity after refeeding, majority of observed signals were variable and the overall tendency was not clear enough to pass the statistical analysis (as attached below).

Effect of sugar refeeding on SDGs neuronal activity visualized by CaLexA

Left, representative images showing cell bodies of SDGs expressing the CaLexA reporter (green) and CsChrimson::mCherry as a marker (magenta).

Right, quantification of the signal intensity. Larvae were either starved for 12 h ("starved") or refed with sucrose after the 12-h starvation ("refed"). Ratio of two fluorescence intensities ($F_{\text{CaLexA}}/F_{\text{mCherry}}$) was normalized to the median value of the starved group. The thick line and the thin error bar represent the median and interquartile range, respectively. (n) indicates the number of cell body observed. The p-value was calculated by Mann–Whitney U-test.

Moreover, the red-shifted CAMPARI signal was not clearly detected in refed larval samples (as attached below), even with the previously reported anti-CAMPARI-red antibody (Moeyaert B, *et al.* 2018 *Nat Commun*; Oikawa I, *et al.* 2023 *eLife*).

CAMPARI2 signal of SDGs in sugar-refed larvae

Immunostaining images showing cell bodies of SDGs expressing the CAMPARI2 reporter. After 12 h of starvation, larvae were refed with sucrose under UV illumination for 1 h. Immunostaining was performed according to the previous reports (Moeyaert B, *et al.* 2018 *Nat Commun*; Oikawa I, *et al.* 2023 *eLife*). Total CAMPARI::HA or the photoconverted CAMPARI-red signals were detected by anti-HA (top) or anti-CAMPARI-red antibody (bottom), respectively.

Although UV is essential for triggering the green-to-red conversion of CAMPARI2, we suspected that sustained UV illumination during the 1-h refeeding may be noxious to larvae and affect the feeding behavior itself. Indeed, when a red dye was added to the sucrose solution to monitor larval feeding, UV illumination during the 1-h sugar treatment strongly suppressed refeeding in majority of starved larvae, as qualitatively judged from their body color (as attached below). This observation led us to conclude that, unfortunately, CAMPARI2 is not suited to study SDGs neuronal activities in the context of sugar-refeeding.

Suppressive effect of UV illumination on sugar-refeeding behavior

After 12 h of starvation, larvae were soaked in a red dye-containing sucrose solution under dark (left) or UV illumination (right) for 1 h. Collected larvae were directly photographed.

Despite our best efforts, it was technically difficult to observe the sugar-induced SDGs activation by methods other than GCaMP. If this reviewer's suggestion stems largely from concerns that InR manipulations may have developmental effects, we can confidently overcome this issue through temporal control of InR activity as discussed below.

(iii) To manipulate InR activity for a limited amount of time, we turned to the temperature sensitive variant of GAL80 (a repressor of GAL4) ubiquitously expressed under the tubulin promoter (*tub-GAL80^{ts}*). Since GAL80 cannot bind to split GAL4s, we turned to *R16C06-GAL4*, the original line found in our screen that cover the neuronal subpopulation including SDGs (**Extended Data Fig. 1**)

For temporal InR inactivation via *tub-GAL80^{ts}*, *R16C06-GAL4*-driven expression of the

dominant negative form of InR (InR^{DN}) was suppressed by rearing larvae at low temperature (21°C) until the last ~7 h preceding behavior assays. Temperature was then shifted to 30°C for GAL80^{ts} inactivation during the 6-h starvation followed by 1 h of sugar refeeding. As a result, the refed test group showed a shortened latency of thermo-nociceptive rolling compared to the refed no-GAL4 control (**Extended Data Fig. 5b**). This effect was temperature dependent, as both groups exhibited comparable levels of refeeding-induced extension in rolling latency when larvae were kept constantly at 21°C (**Extended Data Fig. 5b**). Similarly for conditional InR activation, larvae harboring *UAS-InR^{CA}* (the active form of InR) with *tub-GAL80^{ts}* were reared at 21°C from the day of egg laying to suppress the *R16C06-GAL4*-driven expression through most of the developmental period. Transiently exposing larvae to 30°C caused the delay in thermo-nociceptive rolling onset, whereas rearing them at 21°C without the temperature shift did not (**Extended Data Fig. 5c**).

Collectively, we demonstrated that InR manipulations for only a limited period still induced behavioral phenotypes consistent with those observed without time restriction. This supports our model that, albeit its potential effects on animal development, InR plays a key role in SDGs mediating the sugar-dependent escape modulation. We added the *tub-GAL80^{ts}* experiments to the revised manuscript (p. 19–20, lines 431–448) as **Extended Data Fig. 5b** and **5c**.

3

Third, it is not clear what activates SDGs in regular (non-starved) flies. SDGs must be active under regular state, else the authors would not have seen the effects of silencing SDGs as described on Figure 1-4. The signal(s) that activates SDGs in regular state must be conveyed to the SDGs differently from the one activated by refeeding as inactivating CNs or ILP2 neurons did not appear to affect fed flies' response to nociceptive stimuli (Figure 6-7). Can the authors inspect the connectome to provide some clues as to what are the neurons that directly synapse onto SDGs?

The reviewer brought up a very important point regarding the missing link in the C4da-to-SDGs signaling pathway. Under normal conditions, the nociceptive information should flow first from C4da to SDGs, and then from SDGs to C4da to form the negative feedback loop. Although we had verified the former by Ca²⁺ imaging (**Fig. 4g**), the neural component mediating this flow of information had not been characterized in our study. One such candidate is A08n, a second-order neuron onto which C4da synapses, located

at the VNC in close proximity to SDGs as well as C4da (Hu C, *et al.* 2017 *Nat Neurosci*). If A08n mediates the C4da-to-SDGs signaling, artificial activation of A08n could evoke Ca^{2+} response in SDGs. To test this possibility, we performed Ca^{2+} imaging on RGECO-labeled SDGs while optogenetically stimulating ChR2-expressing A08n. As a result, the peak $\Delta F/F_0$ in SDGs hardly changed upon optogenetic stimulation of A08n, showing no significant difference from larval samples prepared without ATR (as attached below). This suggests that A08n has only minor, if any, contribution to the C4da-to-SDGs signaling. Further physiological studies involving other second-order neurons of C4da may provide a better understanding of the precise neural circuitry for SDGs activation. Since it is still not easy to narrow down the candidates from the recently published larval connectome (Winding M, *et al.* 2023 *Science*), we wish to leave this preliminary data and accompanying discussion to future studies.

Effect of A08n optogenetic stimulation on SDGs neuronal activity.

Left, a schematic of Ca^{2+} imaging in SDGs while optogenetically stimulating A08n neurons.

Right, traces of $\Delta F/F_0$ were measured in larvae reared without ATR (grey) or with ATR (magenta).

The blue-shaded box indicates the 3-s optogenetic stimulation period. Box plots show the median with interquartile range, and whiskers represent the minimum-to-maximum range.

n.s. $p \geq 0.05$ (Mann–Whitney U-test).

Lastly, it might be worth discussing if the suppression of nociceptive response the authors observed the truly the results of animals "prioritizing" feeding over pain or, alternatively, perhaps bingeing on sugar following starvation induces a state (e.g., sleep state) that suppresses movements in general. (One suppose if animals were to prioritize feeding when starved, the impact of nocifensive suppression should be present immediately upon contact with sugars as opposed to waiting for sugars to tripper ILP2 release.) On a related note, it might be interesting to test whether the nociception suppression upon refeeding of starved flies is specific to nutritive sugar, or could similar effects be observed when starved animals are refed with proteins and/or fats.

We thank the reviewer's constructive and interesting suggestions.

(i) We found that sugar-refeeding did not affect larval crawling speed (**Extended Data Fig. 4a**), suggesting that the behavioral phenotypes observed in refed larvae were less likely attributed to general locomotion defects. There remains the interesting possibility that sugar refeeding might affect the "sleep state" of larvae, but we feel setting up the quantification system for larval sleep that strictly meet multiple behavioral criteria (Szuperak M, *et al.* 2018 *eLife*) requires laborious efforts beyond the scope of the present study. At least we added the locomotion data to the revised manuscript (p. 14, lines 308–310) as **Extended Data Fig. 4a**.

(ii) According to the comment, we tested yeast refeeding and observed a modest but statistically significant change in the thermo-nociceptive rolling latency (as attached below). Although this phenomenon is of interest, the underlying mechanism may or may not be the same as the sugar refeeding-induced nociceptive suppression. Rather than diversifying but to gather the main focus on one nutritive condition (*i.e.*, sugar refeeding) and related molecular and neuronal pathways (*i.e.*, insulin/InR and glucose-sensing/insulin-producing neurons) in the present manuscript, we wish to leave this preliminary data for our forthcoming study.

Yeast refeeding causes a modest delay in thermonociceptive rolling onset.

After 6 h of starvation, larvae were refed yeast paste or sucrose for 1 h prior to heat assay. Rolling latency after the heat probe attachment was plotted. The thick line and the thin error bar represent the median and the interquartile range, respectively. a–c $p < 0.005$ (One-way ANOVA Kruskal–Wallis test with Dunn's multiple comparisons test).

Reviewer #3 (Remarks to the Author):

The manuscript by Dojo et al is a stunning manuscript showing a descending nociceptive inhibitory circuit from the brain directly to pain sensing neurons in Drosophila. This cascade is initiated through sugar sensing during the refeeding period, including molecular interactions in the fly brain that initiate the descending circuit. The authors use an array of behavioral, molecular, genetics, and imaging techniques with easy presentation and robust quantification to really tell a complete and impactful story. This work represents an important new addition to the emerging body of literature of descending pain inhibition across a wide variety of animals. This is one of the best manuscripts I have read in a long time and I cannot find any flaws in the work. One potential limitation is there is only a singular read-out for nociception, which is rolling behavior. I understand that this might be a limitation of the system and the relatively limited repertoire of behaviors that a fly larva can display. Perhaps adding place aversion experiments would be nice as another independent readout, but this is not necessary because this paper is already complete. I recommend publication without delay.

Thank you so much for the encouraging comments. We do acknowledge that ideally assessing multiple behavioral readouts would provide deeper mechanistic and ethological insights into the larval nociceptive escape. To this end as suggested, we performed the place aversion test in which larvae were illuminated with blue light, a noxious stimulus that triggers larval avoidance in a C4da-dependent manner (Xiang Y, *et al.* 2010 *Nature*). As attached below, control larvae showed a clear aversion from the area applied with blue light (“+>Kir2.1”, avoidance index = 0.44), whereas genetic silencing of larval C4da via Kir2.1 expression significantly abrogated the photoavoidance (“ppk>Kir2.1”, avoidance index = 0.22). In contrast, SDGs silencing slightly increased the avoidance tendency (“SDGs>Kir2.1”, avoidance index = 0.52), though without statistical significance.

Blue light avoidance of larvae in which C4da or SDGs were genetically silenced.

Left, a schematic of blue light avoidance test. Right, avoidance index measured after 5 min.

Avoidance index was calculated by the following formula: $\frac{\text{(Number of larvae in the non-lighted area)} - \text{(Number of larvae in the lighted area)}}{\text{Total number of larvae}}$. Results were pooled from 2 independent trials each using 12–23 larvae per genotype. (n) represents the total number of larvae tested for each genotype. P values were analyzed by Fisher's exact test.

Further optimization of the light conditions (*e.g.*, intensity and duration) and fine-scale observation of larval escape behaviors at a higher temporal resolution may better clarify the potential impact of SDGs silencing on the larval photoavoidance. We decided to first establish the role of SDGs in nociceptive suppression based on a single yet quantitative index (larval rolling) and leave these additional efforts for expanding the behavioral readout to future studies.

REVIEWERS' COMMENTS

Reviewer #1 (Remarks to the Author):

The authors did a good job addressing my concerns. I now support the publication of this paper in Nature Communications.

Reviewer #2 (Remarks to the Author):

In the revised MS, the authors appropriately addressed the reviewers' concerns. I now support its publication in Nature Communication.

Reviewer #3 (Remarks to the Author):

The authors have done a tremendous job addressing both my comments and those of the other two reviewers. I remain enthusiastic about this paper and would like to see it published.